# DiTAR: Diffusion Transformer Autoregressive Modeling for Speech Generation

**Dongya Jia** [1]   **Zhuo Chen** [1]   **Jiawei Chen** [1]   **Chenpeng Du** [1]   **Jian Wu** [1]   **Jian Cong** [1]   **Xiaobin Zhuang** [1]
**Chumin Li** [1]   **Zhen Wei** [1]   **Yuping Wang** [1]   **Yuxuan Wang** [1]

## Abstract

Several recent studies have attempted to autoregressively generate continuous speech representations without discrete speech tokens by combining diffusion and autoregressive models, yet they often face challenges with excessive computational loads or suboptimal outcomes. In this work, we propose Diffusion Transformer Autoregressive Modeling (DiTAR), a patch-based autoregressive framework combining a language model with a diffusion transformer. This approach significantly enhances the efficacy of autoregressive models for continuous tokens and reduces computational demands. DiTAR utilizes a divide-and-conquer strategy for patch generation, where the language model processes aggregated patch embeddings, and the diffusion transformer subsequently generates the next patch based on the output of the language model. For inference, we propose defining temperature as the time point of introducing noise during the reverse diffusion ODE to balance diversity and determinism. We also show in the extensive scaling analysis that DiTAR has superb scalability. In zero-shot speech generation, DiTAR achieves state-of-the-art performance in robustness, speaker similarity, and naturalness. Audio samples are presented in https://spicyresearch.github.io/ditar.

## 1. Introduction

Recently, the autoregressive language model (LM) has demonstrated strong generative capabilities and scaling properties(Achiam et al., 2023; Touvron et al., 2023; Yang et al., 2024; Team et al., 2023), where discrete tokenization is commonly used. While discretization is natural for text,

[1]ByteDance Seed. Correspondence to: Dongya Jia <jiadongya@bytedance.com>, Zhuo Chen <zhuo.chen1@bytedance.com>.

it is not as straightforward for other modalities, such as images, video, and audio. Due to bitrate limitations, discrete representation often fails to reconstruct complex modalities with high fidelity. In comparison, continuous tokens can better reconstruct the original information(Fan et al., 2024; Rombach et al., 2022). To alleviate this challenge, a two-stage strategy is commonly applied for autoregressive multi-media generation, where a lossy discrete token is first generated by an LM, followed by a token-based diffusion for detail enrichment. However, such a cascaded design suffers from error accumulation and limits the scalability of the language model. Therefore, autoregressive modeling of continuous representations is meaningful.

The diffusion model has been proven effective at modeling continuous representations(Peebles & Xie, 2023; Rombach et al., 2022; Anastassiou et al., 2024), and integrating it with autoregressive models is a major research trend. (Li et al., 2024a; Fan et al., 2024) proposes incorporating a diffusion head into a language model for image generation, pioneering a new approach. However, the performance significantly falls short when a causal-attention language model forms the backbone. Another approach, such as ARDiT(Liu et al., 2024b) or Transfusion(Zhou et al., 2024), repurposes the language model's parameters for diffusion, leading to substantial computational demands. This raises the question of whether we can autoregressively predict continuous tokens while ensuring high-quality generation and maintaining reasonable computational demands.

To tackle this challenge, we introduce the Diffusion Transformer AutoRegressive (DiTAR) modeling, a framework that seamlessly combines transformer diffusion with a language model. We attribute the subpar performance of a language model with a diffusion head to the unidirectional dependency imposed by causal attention, which conflicts with the close inter-frame correlations characteristic of continuous tokens. In DiTAR, we propose a divide-and-conquer strategy that breaks continuous tokens into multiple patches. A language model is responsible for inter-patch prediction, while a diffusion transformer handles intra-patch prediction. Specifically, a diffusion transformer with bidirectional attention (DiT), noted for its state-of-the-art performance across various generative fields(Liu et al., 2024a; Peebles & Xie, 2023; Anastassiou et al., 2024), is employed to predict

localized patches, named LocDiT. Furthermore, we propose a broadly applicable guidance method and introduce historical contexts to LocDiT to further enhance its generative capabilities. After patchification, the causal language model handling long contexts receives shorter sequences, further reducing computational load.

For inference, temperature-based sampling balances exploration and exploitation and is crucial in discrete-valued language models. However, its application in continuous-valued LMs remains underexplored. We define temperature as the point at which noise is introduced along the reverse ODE trajectory and propose a temperature-based sampling method for the continuous-valued AR model. Distinct from previous SDE-based approaches(Dhariwal & Nichol, 2021; Li et al., 2024a) that require numerous steps, our method suits quicker ODE solvers and adeptly balances diversity with determinism.

DiTAR, rooted in a language model, excels at zero-shot generation tasks. We apply DiTAR to zero-shot text-to-speech (TTS) that aims to generate speech from unseen speakers' prompts. Unlike systems(Chen et al., 2024b; 2025; Anastassiou et al., 2024; Du et al., 2024) using a coarse-to-fine pipeline, DiTAR simplifies the process by having the language model directly predict final features, obviating the need for multiple stages. It achieves state-of-the-art results in robustness, speaker similarity, and naturalness while demanding far less computational power than competing models.

In summary, our contributions to the community include:

- We introduce DiTAR, a patch-based autoregressive framework that seamlessly blends LM and DiT, maintaining their strengths while offering superb generative capabilities and reduced computational demands.

- In inference, we propose a new definition of temperature and a fast temperature-based sampling method tailored for autoregressive models with diffusion loss.

- We apply DiTAR to zero-shot speech generation and achieve SOTA performance with a much lower computational load.

## 2. Related Works

**Integrating Autoregressive Language Model and Diffusion.** Language models are primarily used for discrete representations, while diffusion excels in modeling continuous distributions. Integrating them for multimodal modeling is a crucial research direction. Some efforts (Wu et al., 2023; Liu et al., 2024b; Chen et al., 2024a) enable diffusion to have autoregressive capabilities by varying the denoising rates between consecutive tokens to achieve earlier predictions for preceding tokens. Transfusion(Zhou et al., 2024) utilizes a shared transformer for both diffusion and language models, employing causal attention for discrete tokens and bidirectional attention for continuous tokens. However, it does not naturally support the autoregressive generation of continuous tokens. These approaches repurpose language model parameters for diffusion, which significantly increases computational demands as the sequence lengthens and the language model size grows. Most relevant to our work, Li et al. (2024a) proposes a diffusion head for next-token prediction. However, its application in causal language models results in relatively poor performance.

**Patchification in Generative Modeling.** In speech(Wang et al., 2017; Meng et al., 2024; Chen et al., 2024b), image(Peebles & Xie, 2023; Li et al., 2024b), and video generation(Liu et al., 2024a), the patchification technique is widely applied. In these works, patchification primarily aims to reduce the computational load by shortening the sequence length. However, in this paper, patchification not only lowers computational demands but also enables bidirectional modeling on patches within our autoregressive framework, further improving modeling effectiveness.

**Zero-Shot Text-to-Speech.** Zero-shot TTS aims to generate speech with prompts from unseen speakers. Existing works can be divided into two categories: multi-stage and single-stage. Multi-stage approaches(Chen et al., 2024b; 2025; Lee et al., 2023; Jiang et al., 2023b;a; Anastassiou et al., 2024), represent speech using various representations, primarily divided into coarse and fine categories. The autoregressive language model is often adopted to predict the coarse representations, usually in discrete values and low information, such as semantics(Kharitonov et al., 2023; Anastassiou et al., 2024; Hsu et al., 2021; Chung et al., 2021; Baevski et al., 2020) and prosody(Jiang et al., 2023b;a; Ju et al., 2024). And then another model is adopted to conduct coarse-to-fine. Single-stage methods focus on generating high-information continuous representations, such as Mel spectrograms or latents of auto-encoders, which can directly reconstruct audio waveforms. Diffusion based on the non-causal attention transformer is the primary method employed(Chen et al., 2024c; Eskimez et al., 2024; Gao et al., 2023; Le et al., 2024). Additionally, some approaches(Meng et al., 2024; Wang et al., 2017; Shen et al., 2018; Li et al., 2019) directly use AR to model Mel-spectrograms of speech but often involve using Dropout(Srivastava et al., 2014) at the input stage to guarantee generation robustness, resulting in weaker in-context learning capabilities. These methods are better suited for in-set speaker TTS. This paper introduces a single-stage autoregressive method for zero-shot TTS, achieving SOTA generation robustness and voice similarity.

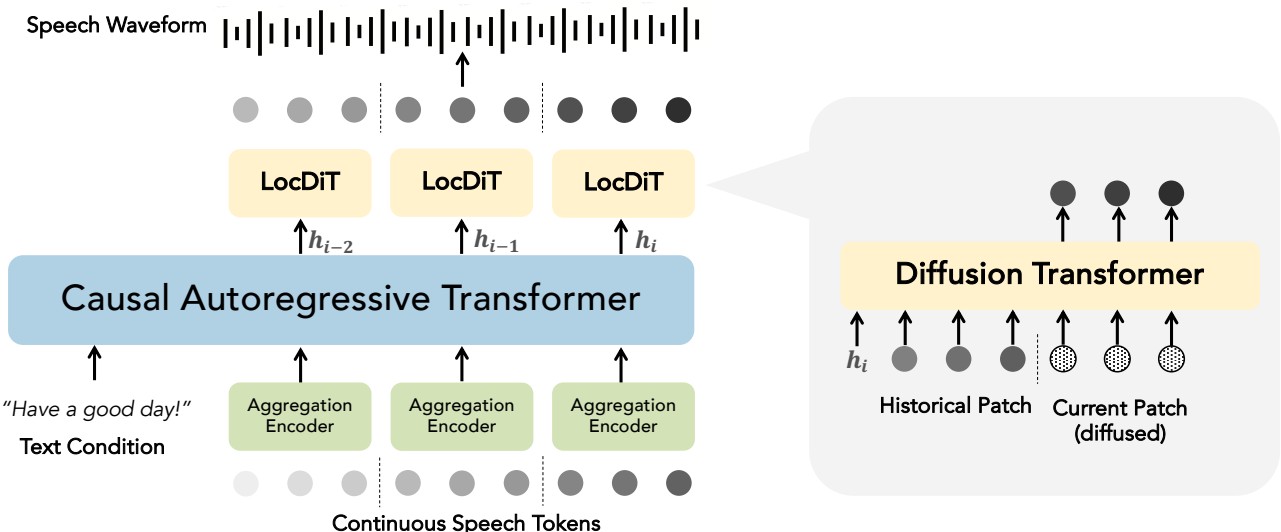

Figure 1: DiTAR is composed of an aggregation encoder for input, a causal language model backbone, and a diffusion decoder, LocDiT, predicting local patches of tokens.

## 3. Method

In a nutshell, we propose DiTAR, a patch-based autoregressive system based on continuous representation. This system amalgamates the strengths of the causal-attention AR and bidirectional-attention transformer diffusion.

### 3.1. Overview

#### 3.1.1. FORMULATION

DiTAR is an autoregressive model via next-token prediction. Consider a sequence of continuous tokens $\boldsymbol{x} = (\boldsymbol{x}_1, \boldsymbol{x}_2, ..., \boldsymbol{x}_N)$, we can factorize the joint distribution of the sequence by the chain rule of probability:

$$p_\theta(\boldsymbol{x}_1, \boldsymbol{x}_2, ..., \boldsymbol{x}_N) = \prod_{i=1}^{N} p_\theta(\boldsymbol{x}_i|\boldsymbol{x}_1, \boldsymbol{x}_2, ..., \boldsymbol{x}_{i-1}) \quad (1)$$

where $\theta$ denotes the parameters of an AR generative model. Noting the high similarity among adjacent continuous tokens, it is evident that a bidirectional dependency exists within local regions(Tian et al., 2024). Based on this discovery, we aggregate local $\boldsymbol{x}_i$ into patches with a size of $P$, and then employ bidirectional attention to model the tokens inside each patch. We can divide the model into two parts, $\theta_a$ and $\theta_b$: $\theta_a$ denotes the autoregressive model responsible for long context learning via $p_{\theta_a}(\boldsymbol{h}_i|\boldsymbol{x}_1, \boldsymbol{x}_2, ..., \boldsymbol{x}_i)$, while $\theta_b$ denotes a bidirectional-attention diffusion transformer executing next-patch prediction via $p_{\theta_b}(\boldsymbol{x}_{i+1}, ..., \boldsymbol{x}_{i+P}|\boldsymbol{h}_i)$, where $\boldsymbol{h}_i$ is the output of language model and condition for diffusion.

We validate the effectiveness of DiTAR on the zero-shot text-to-speech task. Based on the formulation, we regard zero-shot TTS as a conditional continuation task for the AR model like Chen et al. (2024b), where prompting texts, target text, and prompting speech are concatenated and fed into the model as prefix context, then the model autoregressively generates the target speech given the context.

#### 3.1.2. OVERALL ARCHITECTURE

Li et al. (2024a) finds that the performance of the causal-attention autoregressive model combined with diffusion loss is significantly inferior to systems utilizing full attention. To address the issue, we propose a divide-and-conquer strategy, where a long sequence of continuous tokens is divided into multiple patches. A language model is responsible for inter-patch prediction, while a diffusion transformer handles intra-patch prediction. As shown in Figure 1, the backbone of our system is a causal-attention transformer with next-token prediction. Each patch of continuous tokens is processed with an aggregation encoder into a single vector, which is then fed into the AR model to get the output embedding $\boldsymbol{h}_t$. $\boldsymbol{h}_t$ serves as the condition of the following diffusion decoder, LocDiT. Following (Li et al., 2024a), a diffusion loss is used for the output continuous tokens at training time.

### 3.2. LocDiT: Next-Patch Bidirectional Modeling

The diffusion transformer(Peebles & Xie, 2023; Liu et al., 2024a) has achieved success across numerous generative fields. These approaches leverage the full receptive field of the bidirectional transformer to generate entire samples. In our work, we propose using a bidirectional transformer diffusion, called Local Diffusion Transformer (LocDiT), to generate localized continuous token patches.

LocDiT generates the next patch of speech given the AR's output. However, we have found that diffusion struggles to predict the next patch under these conditions. To capitalize on the context-learning potential of the diffusion transformer, we propose a context-aware diffusion approach. Specifically, as illustrated in the right half of Fig 1, historical patches of tokens are utilized as prefix inputs for the LocDiT, thereby aligning the task more closely with outpainting and significantly improving the generation performance. Detailed quantitative results are discussed in Section 4.3.

Furthermore, the design also implies an inherent coarse-to-fine manner. Current approaches(Anastassiou et al., 2024; Du et al., 2024; Chen et al., 2024b) commonly explicitly delineate both coarse and fine features: a language model typically predicts the coarse feature, while another network achieves the transition from coarse to fine. However, multi-stage methods are prone to cumulative errors. In contrast, DiTAR functions in a seamless end-to-end manner, condensing each patch of tokens into an implicit coarse feature space that LocDiT subsequently expands into high-fidelity continuous tokens.

### 3.3. LM Guidance

Classifier-free guidance (CFG)(Ho & Salimans, 2022) is extensively employed to enhance the condition adherence of generative models. In diffusion, the unconditional and conditional models share parameters and are jointly trained by intermittently omitting the condition during training. At inference, the outputs from these two models are merged with a parameter $w$ balancing the trade-off between diversity and fidelity. This is equivalent to sampling under such a distribution $\tilde{p}_\theta(\boldsymbol{z}_t, c) \propto p_\theta(\boldsymbol{z}_t|c)p_\theta(c|\boldsymbol{z}_t)^w$, where $\theta$ denotes the model parameters, $c$ denotes the condition and $\boldsymbol{z}_t$ denotes the noisy sample at the time of $t$. For discrete-valued language models, classifier-free guidance(Sanchez et al., 2023) often involves computing the language model twice to obtain conditional and unconditional logits, which can be computationally expensive.

For DiTAR, a model that incorporates both an LM and a diffusion head, we propose LM guidance, an effective approach that requires only two computations of the diffusion head and one computation of the LM. Specically, the $i$th output $\boldsymbol{h}_i$ of the LM essentially represents all historical inputs $(\boldsymbol{x}_1, \boldsymbol{x}_2, ..., \boldsymbol{x}_i)$ and we randomly replace the $\boldsymbol{h}_i$ with a dummy embedding $\boldsymbol{h}_\varnothing$ along the sequence in training. At inference time, we samples from the distribution $p_\theta(\boldsymbol{z}_{i,t}|\boldsymbol{x}_1, ..., \boldsymbol{x}_{i-1})p_\theta(\boldsymbol{x}_1, ..., \boldsymbol{x}_{i-1}|\boldsymbol{z}_{i,t})^w$ by:

$$\tilde{\boldsymbol{\epsilon}}_\theta(\boldsymbol{z}_{i,t}, \boldsymbol{h}_i) = (1+w)\boldsymbol{\epsilon}_\theta(\boldsymbol{z}_{i,t}, \boldsymbol{h}_i) - w\boldsymbol{\epsilon}_\theta(\boldsymbol{z}_{i,t}, \boldsymbol{h}_\varnothing) \quad (2)$$

where $\boldsymbol{z}_{i,t}$ denotes the $i$th noisy sample in the sequence at

the time of $t$ for diffusion, $\boldsymbol{\epsilon}_\theta$ denotes the score estimated by the LocDiT, $\boldsymbol{h}_\varnothing$ denotes the dummy vector representing unconditional modeling. Operating in the velocity space with a conditional flow-matching target is also equivalent. We show that the approach significantly enhances the model's performance in Section 4.3.

### 3.4. Temperature for Continuous-Valued LMs

Temperature-based sampling, which strikes a balance between exploration and exploitation, is fundamental in discrete-valued LMs. However, its application in continuous-valued LMs has been less extensively studied. Li et al. (2024a) proposes a sampling method based on the DDPM reversed process(Ho et al., 2020), which is a first-order SDE solver, where the temperature is defined as the noise scaling factor at each step. Due to the stochastic nature of the Wiener process, SDE solvers require small step sizes, necessitating a higher number of steps for convergence(Song et al., 2020b). Additionally, this concept of temperature cannot be readily extended to the more widely used and faster ODE solvers prevalent in contemporary applications(Lu et al., 2022a;b; Song et al., 2020a). To address these issues, we propose a new sampling method with a new definition of temperature, which is compatible with the ODE solvers.

We define the temperature $\tau \in [0, 1]$ as *the time point to introduce noise* while solving the reverse ODE of diffusion. Specifically, consider a Gaussian diffusion forward process defined on per-patch by $\boldsymbol{x}_t = \alpha_t \boldsymbol{x}_0 + \sigma_t \boldsymbol{\varepsilon}$, where $\boldsymbol{x}_0 \sim q_{data}(\boldsymbol{x}_0)$, $\boldsymbol{\varepsilon} \sim \mathcal{N}(\boldsymbol{0}, \boldsymbol{I})$, $t \in [0, 1]$, $\alpha_t$ and $\sigma_t$ collectively defines the flow path. At $\tau = 1$, the sampling process is equivalent to the standard ODE sampling process, where we solve the reverse ODE $d\boldsymbol{x}_t = \boldsymbol{v}_\theta(\boldsymbol{x}_t, t)dt$ from 1 to 0, where $\boldsymbol{x}_1 \sim \mathcal{N}(\boldsymbol{0}, \boldsymbol{I})$ and $\boldsymbol{v}_\theta$ denotes predicted vector field. At $\tau = 0$, no random noise is introduced, leading to a completely deterministic process. Considering that 0 is the value with the highest likelihood in the standard Gaussian distribution, we define the greedy sampling by sampling from $x_1 \equiv \boldsymbol{0}$, thus ensuring determinism.

When $0 < \tau < 1$, we introduce random noise at $\tau$ by using the forward process to diffuse an estimated $\boldsymbol{x}_0$. Eq 3 and 4 summarize the iterative process, where the Euler method is used as the default ODE solver for illustration.

$$\boldsymbol{x}_1 \sim \begin{cases} \mathcal{N}(\boldsymbol{0}, \boldsymbol{I}) & \text{if } \tau = 1 \\ \boldsymbol{0} & \text{if } 0 \leq \tau < 1 \end{cases} \quad (3)$$

$$\boldsymbol{x}_t = \begin{cases} \boldsymbol{x}_{t+\Delta t} - \boldsymbol{v}_\theta(\boldsymbol{x}_{t+\Delta t}, t + \Delta t)\Delta t & \text{if } t \neq \tau \\ \alpha_t \boldsymbol{x}_\theta(\boldsymbol{x}_{t+\Delta t}, t + \Delta t) + \sigma_t \boldsymbol{\varepsilon} & \text{if } t = \tau \end{cases} \quad (4)$$

where $\boldsymbol{x}_\theta$ represents the estimated $\boldsymbol{x}_0$, which can be de-

rived from the estimated velocity or score through a linear transformation. The derivation process is detailed in Appendix A.3. In Section 4.3, we demonstrate that $\tau$ effectively balances diversity and stability. The sampling process is summarized in Algorithm 1.

---

**Algorithm 1** Temperature sampling

---

**Input:** $v$-prediction model $\boldsymbol{f_\theta}(.,.)$, discretized time points $t_1 < t_2 < ... < t_{N-1} \in [0,1)$, $t_N = 1$, ODE solver $\boldsymbol{\Psi}(.,.,.)$, transformation function $\mathcal{F}$, temperature $\tau$

$\eta \leftarrow \underset{n=1,2,...,N}{\text{argmin}} |t_n - \tau|$

**if** $\eta = N$ **then**

  Sample initial noise $\boldsymbol{x}_{t_N} \sim \mathcal{N}(\boldsymbol{0}, \boldsymbol{I})$

**else**

  initial noise $\boldsymbol{x}_{t_N} \leftarrow \boldsymbol{0}$

**end if**

**for** $n = N - 1$ **to** 1 **do**

  $\hat{\boldsymbol{v}} \leftarrow \boldsymbol{f_\theta}(\boldsymbol{x}_{t_N}, t_N)$

  **if** $t_n = \eta$ **then**

    $\hat{\boldsymbol{x}} \leftarrow \mathcal{F}(\hat{\boldsymbol{v}})$

    Sample $\boldsymbol{z} \sim \mathcal{N}(\boldsymbol{0}, \boldsymbol{I})$

    $\boldsymbol{x} \leftarrow \alpha_{t_{n+1}}\hat{\boldsymbol{x}} + \beta_{t_{n+1}}\boldsymbol{z}$

  **else**

    $\boldsymbol{x} \leftarrow \boldsymbol{\Psi}(\hat{\boldsymbol{v}}, n+1, n)$

  **end if**

**end for**

**Output:** $\boldsymbol{x}$

---

## 3.5. Implemenations

### 3.5.1. CONTINUOUS SPEECH TOKENIZATION

Following LDM(Rombach et al., 2022), we use a variational auto-encoder (VAE)(Kingma, 2013) to convert the waveform into the distribution of latent $z$, represented by mean and variance. The encoder of the VAE consists of multiple layers of the convolutional network, and the decoder's architecture follows BigVGAN(Lee et al., 2022). The adversarial training scheme also follows Lee et al. (2022). We adopt the multi-period discriminator (MPD) and multi-scale discriminator (MSD) proposed by Kong et al. (2020) as our discriminators. In our setup, the 24000hz waveform is compressed into 40Hz latent with a dimension of 64.

### 3.5.2. MODEL

DiTAR consists of three modules: aggregation encoder, language model, and decoder (LocDiT). In our implementation, all of them are based on a transformer architecture. Specifically, both the encoder and decoder employ bidirectional attention masks, while the LM utilizes a causal attention mask. All transformers adopt the Pre-Norm(Xiong et al., 2020) architecture and utilize RMSNorm(Zhang & Sennrich, 2019) and RoPE (Su et al., 2024). Each patch of continuous

tokens, together with a learnable special token positioned at the beginning of the sequence similar to (Devlin, 2018), is fed into the aggregation encoder. The output corresponding to the special token's position serves as the aggregation embedding. Aggregation embeddings from different patches form a sequence that is processed by the LM. In LocDiT, LM's outputs and the time embedding are added, along with historical context patches and noisy target tokens, forming a new sequence that serves as the input of LocDiT. When calculating the loss, we only consider the output corresponding to the position of noisy target tokens. During training, LM's output is randomly replaced by a vector of all zeros with a probability of 0.1 to enable LM guidance for LocDiT.

### 3.5.3. DIFFUSION FORMULATION

Following Song et al. (2020b); Lu & Song (2024), we adopt a variance-preserving diffusion process defined by

$$\boldsymbol{x}_t = \alpha_t \boldsymbol{x}_0 + \sigma_t \boldsymbol{\varepsilon} \tag{5}$$

$$= \cos\left(\frac{\pi t}{2}\right)\boldsymbol{x}_0 + \sin\left(\frac{\pi t}{2}\right)\boldsymbol{\varepsilon} \tag{6}$$

where $\boldsymbol{x}_0 \sim q(\boldsymbol{x}_0)$ denotes the data, $\boldsymbol{\varepsilon} \sim \mathcal{N}(\boldsymbol{0}, \boldsymbol{I})$ denotes the standard Gaussian noise, $t \in [0,1]$. We employ a conditional flow-matching loss(Lipman et al., 2022):

$$L_{diff} = \mathbb{E}_{t,\boldsymbol{x}_0,\boldsymbol{\varepsilon}}\left[\|\boldsymbol{v}_\theta(\boldsymbol{x}_t, t) - \boldsymbol{v}(\boldsymbol{x}_t, t)\|_2^2\right] \tag{7}$$

where the velocity is defined as: $\boldsymbol{v}(\boldsymbol{x}_t, t) = \dot{\boldsymbol{x}}_t = \dot{\alpha}_t \boldsymbol{x}_t + \dot{\sigma}_t \boldsymbol{\varepsilon}$. At inference time, We employed the DDIM sampler(Song et al., 2020a), which is essentially an Euler ODE sampler with respect to signal-to-noise ratio $\frac{\alpha_t^2}{\sigma_t^2}$ instead of $t$, proved to better align with the semi-linear property of the diffusion ODE(Lu et al., 2022a).

### 3.5.4. ZERO-SHOT TTS SYSTEM

The text sequence is converted into phonemes and processed through a lookup table to obtain text embeddings. Speech tokens are processed by the aggregation encoder to produce speech embeddings, which are then concatenated with the text embeddings. The embedding sequence serves as the input of the LM of DiTAR. During training, the loss of the text is not included. Additionally, we introduce a binary classifier with a fully connected layer at LM's output to predict when to stop following Wang et al. (2017); Li et al. (2019). The loss function for zero-shot TTS can be summarized as $L = L_{diff} + L_{stop}$. During inference, text, target text, and prompting audio are fed as prefix input to DiTAR's LM, which then autoregressively generates the target audio.

Table 1: Objective evaluation results of DiTAR and other systems on two subsets of LibriSpeech test-clean. Specifically, subset A is used and reported by NaturalSpeech3, while subset B is released by F5TTS. ♦ denotes the scores reported in NaturalSpeech3. ♣ means the results obtained from the authors of F5TTS. ♠ means the results are obtained via released checkpoints. The boldface and underline indicate the best and the second-best result, respectively. ↑ and ↓ indicate lower or higher values are better. Abbreviation: Disc.(discrete), Cont.(Continuous), AR(autoregressive model), NAR(non-autoregressive model), NFE(number of function evaluation)

| Type | System | #Params | Training Data | WER(%)↓ | SIM↑ | UTMOS↑ | TFLOPs↓ |
|---|---|---|---|---|---|---|---|
| **LibriSpeech test-clean A** | | | | | | | |
| - | Human | - | - | 0.34 | 0.68 | 4.14 | - |
| - | Vocoder | - | - | 0.34 | 0.63 | 4.03 | - |
| Disc. AR + Disc. NAR | VALL-E ♦ | 0.4B | Librilight | 6.11 | 0.47 | 3.68 | ∼ 2.99 |
| Disc. AR + Cont. NAR | MegaTTS 2 ♦ | 0.5B | Librilight | 2.32 | 0.53 | 4.02 | ∼ 0.06 |
| Disc. NARs | NaturalSpeech 3 ♦ | 0.5B | Librilight | 1.81 | **0.67** | **4.30** | ∼ 8.92 |
| Cont. NAR | NaturalSpeech 2 ♦ | 0.4B | Librilight | 1.94 | 0.55 | 3.88 | ∼ 12.89 |
| Cont. NAR | Voicebox (NFE=32) ♦ | 0.4B | Librilight | 2.14 | 0.48 | 3.73 | ∼ 60.89 |
| Cont. AR | DiTAR (NFE=10) | 0.6B | Librilight | **1.78** | 0.64 | 4.15 | ∼ 2.75 |
| **LibriSpeech test-clean B** | | | | | | | |
| - | Human | - | - | 2.23 | 0.69 | 4.10 | - |
| - | Vocoder resynthesized | - | - | 2.38 | 0.66 | 3.97 | - |
| Disc. NARs | MaskGCT (NFE=50) ♠ | 1.1B | Emilia | 2.72 | **0.69** | 3.90 | ∼ 116.66 |
| Cont. NAR | E2TTS (NFE=32) ♣ | 0.3B | Emilia | 2.95 | **0.69** | 3.56 | ∼ 56.46 |
| Cont. NAR | F5TTS (NFE=32) ♣ | 0.3B | Emilia | 2.42 | 0.66 | 3.88 | ∼ 37.36 |
| Cont. AR | DiTAR (NFE=10) | 0.6B | Emilia | **2.39** | 0.67 | **4.22** | ∼ 2.75 |

Table 2: Subjective evaluation results on LibriSpeech test-clean subset B. We compare DiTAR with several leading NAR systems.

| System | N-MOS | Q-MOS | S-MOS | CMOS |
|---|---|---|---|---|
| Human | 3.89 | 3.61 | 3.56 | +0.18 |
| E2TTS | 3.27 | 3.44 | 3.15 | -0.32 |
| F5TTS | 3.36 | 3.58 | 3.33 | -0.04 |
| DiTAR | **3.69** | **3.87** | **3.55** | **0.00** |

## 4. Experiments

### 4.1. SOTA Performance in Zero-Shot TTS

In this subsection, we benchmark DiTAR against leading systems and demonstrate its state-of-the-art performance.

#### 4.1.1. SETUP

To ensure a fair evaluation of zero-shot TTS, it is essential to consider prompt audio, texts, and tools. We standardize these variables to facilitate a more objective and fair comparison between systems.

**Training and Evaluation Dataset.** We consider two open-source datasets as our training dataset. 1) Librilight(Kahn et al., 2020), containing 60K hours of English speech data from LibriVox audiobooks. 2) Emilia(He et al., 2024), a multilingual dataset containing around 100k hours of speech.

We adopt three open-source datasets for evaluation: 1) Lib-

riSpeech(PC)(Panayotov et al., 2015; Meister et al., 2023) test-clean, containing 40 distinct English speakers and a 5.4-hour speech. We employ two established subsets: subset A from NaturalSpeech3, featuring 40 three-second speech prompts and 40 target samples, and subset B from F5TTS, which includes 40 prompts and 1127 samples. 2) Seed-ZH: a subset from DiDiSpeech 2(Guo et al., 2021), a Chinese speech dataset, containing 1088 prompts and targets. 3) Seed-EN: a subset from Common Voice(Ardila et al., 2019), a crowdsourcing English speech dataset with diverse accents, containing 2020 prompts and targets.

**Evaluation Metrics.** We evaluate four objective metrics: 1) Word Error Rate (WER), which assesses generation robustness. For consistency, we use the same ASR setups as previous studies to transcribe generated speech across different test sets. Specifically, we adopt a Hubert-based model[1] and Faster-whisper-large-v3 [2](Radford et al., 2022) for the subset A and B of Librispeech test-clean, respectively, Whisper-large-v3[3] for Seed-EN and Paraformer-zh for Seed-ZH. 2) Speaker similarity to the prompt audio (SIM). Specifically, we employed WavLM-large(Chen et al., 2022) to compute the cosine distance between the generated and reference speech. 3) UTMOS(Saeki et al., 2022), an automatic predicted mean opinion score (MOS) to evaluate speech quality. 4) Floating point operations (FLOPs), mea-

---

[1]https://huggingface.co/facebook/hubert-large-ls960-ft
[2]https://huggingface.co/Systran/faster-whisper-large-v3,version:0.10.1
[3]https://huggingface.co/openai/whisper-large-v3

suring the computational load of models. The calculation process is detailed in Appendix A.4.

For subjective evaluation, we employ four MOS metrics: 1) N-MOS for naturalness, 2) Q-MOS for sound quality, 3) S-MOS for speaker voice similarity 4) CMOS for side-by-side comparison with human audios.

**Model Setup and Baselines.** We benchmark DiTAR against diverse zero-shot TTS systems with varying architectures, including both multi-stage and single-stage schemes that operate in discrete or continuous value spaces. We conduct comparisons using DiTAR with 0.6 billion parameters and a patch size of 4. During inference, DiTAR's LocDiT uses an NFE (Number of Function Evaluations) of 10. Specific details about the parameters of DiTAR are provided in Appendix A.2.

### 4.1.2. EXPERIMENTAL RESULTS

We conduct a multi-dimensional comparison of DiTAR with other baseline works. For objective metrics, Table 1 presents the evaluation results on LibriSpeech test-clean. For subjective evaluation, we invite 10 English experts to rate the generated audio. For N-MOS, Q-MOS, and S-MOS metrics, the generated audios are rated on a scale of 1 to 5. For CMOS, experts compare the generated audio against the ground truth (GT) and assign scores from -2 to 2. Subjective results are detailed in Table 2.

**Generation Robustness.** As shown in Table 1, under two different training data configurations and test sets, DiTAR consistently delivered the best WER, showcasing robust synthesis performance matched by NAR systems with phone-level duration models. Table 3 further details DiTAR's comparison with models trained on proprietary data, highlighting its superior synthesis stability.

**Speaker Similarity.** We perform both objective and subjective assessments of speaker similarity. Objectively, as Table 1 illustrates, DiTAR delivers strong SIM scores, on par with NAR systems. Subjectively, as detailed in Table 2, DiTAR outperforms leading NAR systems in S-MOS scores, demonstrating excellent in-context learning capabilities.

**Naturalness.** We assess speech naturalness using the subjective metrics N-MOS and CMOS. As detailed in Table 2, DiTAR excels over leading NAR systems, achieving the highest scores for naturalness.

**Audio Quality.** We use the objective metric UTMOS and the subjective metric Q-MOS to evaluate audio quality. Table 1 demonstrates that DiTAR achieves competitive UTMOS scores on the LibriSpeech test-clean dataset, ranking second only to NaturalSpeech3 in subset A while outperforming all other systems in subset B.. Subjectively, as shown in Table 2, DiTAR exceeds even the ground truth

Table 3: Objective evaluation results of DiTAR and various systems on Seed-EN and Seed-ZH.

| System | Seed-EN | | Seed-ZH | |
|---|---|---|---|---|
| | WER(%)↓ | SIM↑ | WER(%)↓ | SIM↑ |
| Human | 2.06 | 0.73 | 1.254 | 0.750 |
| Seed-TTS$_{DiT}$ | 1.733 | **0.790** | 1.178 | **0.809** |
| CosyVoice | 4.29 | 0.609 | 3.63 | 0.723 |
| CosyVoice 2 | 2.57 | 0.652 | 1.45 | 0.748 |
| CosyVoice 2-S | 2.38 | 0.654 | 1.45 | 0.753 |
| FireRedTTS | 3.82 | 0.46 | 1.51 | 0.63 |
| MaskGCT | 2.623 | 0.717 | 2.273 | 0.774 |
| E2TTS | 2.19 | 0.71 | 1.97 | 0.73 |
| F5TTS | 1.83 | 0.67 | 1.56 | 0.76 |
| DiTAR | **1.685** | 0.735 | **1.023** | 0.753 |

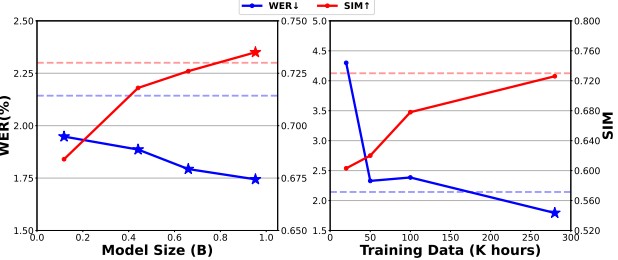

Figure 2: The performance of DiTAR consistently improves with increases in either training data or model size. The star marker indicates performance that surpasses human levels.

in Q-MOS scores, highlighting the superior quality of its outputs.

**Computaional Load.** Table 1 indicates that DiTAR not only ensures high-quality audio generation but also dramatically cuts computational demands by approximately $3 \sim 43\times$ compared to other NAR systems. Although hybrid systems require less computational power, they tend to produce lower-quality outputs. Further details on inference efficiency are discussed in 4.4.2.

### 4.1.3. COMPARISON WITH ADDITIONAL MODELS

To assess the upper-bound performance of DiTAR, we train DiTAR with 1 billion parameters on 280k-hour data and further compare DiTAR against a wider range of models, including some closed-source systems, such as Seed-TTS. We conduct tests on datasets for two languages, namely Seed-EN and Seed-ZH. As shown in Table 3, DiTAR achieves the best generation robustness and excellent speaker similarity. All results of other systems are reported in their papers.

### 4.2. Scaling Behaviors

In this subsection, we explore DiTAR's scaling properties for zero-shot speech generation, focusing on model architecture and training data, using Seed-EN as the test set.

Table 4: Scaling behavior of different components.

| System | WER(%)↓ | SIM↑ |
|---|---|---|
| DiTAR (0.4B) | 1.876 | 0.716 |
| Encoder size ×4 | 1.821 | 0.72 |
| Language model size ×4 | **1.695** | **0.727** |
| LocDiT size ×4 | **1.785** | **0.726** |

Table 5: The impact of the number of historical patches in LocDiT. ⋆ indicates that many samples fail to stop during generation.

| Patch Size | # Historical Patches | WER(%)↓ | SIM↑ |
|---|---|---|---|
| | 2 | 3.334 | 0.716 |
| 1 | 1 | 6.131 | 0.692 |
| | 0 (not used) | 53⋆ | 0.34⋆ |
| | 2 | **1.809** | **0.73** |
| 4 | 1 | **1.736** | **0.72** |
| | 0 (not used) | 22.874⋆ | 0.56⋆ |

**The performance of DiTAR consistently enhances as either data size or model size increases.** We conduct scaling experiments concerning both data and model size. We expand the training data from 20k to 280k hours with a 0.6B model to assess performance changes. For model size, we scale from 0.1B to 1B parameters using 280k hours of training data, simultaneously increasing the parameters of the encoder, language model, and LocDiT. For the detailed setups see Table 8. As shown in Figure 2, the model's WER and SIM consistently improve as training data and model parameters increase.

**The language model and diffusion decoder benefit more from scaling.** We scale the encoder, language model, and decoder (LocDiT) individually to assess their impacts, starting with a 0.4B parameter model trained on 280k hours data . Given that the encoder and decoder process shorter sequences, we allocated more parameters to the language model empirically. Table 4 demonstrates that enlarging the language model and LocDiT enhances performance, whereas increasing the encoder size has little effect on outcomes.

### 4.3. Method Analysis and Ablation Study

In this subsection, we conduct a detailed analysis of the various components of DiTAR. Unless specified otherwise, we default to using DiTAR with 0.6 billion parameters, a patch size of 4, and NFE=10, tested on the Seed-EN dataset.

**Patch Size.** LocDiT utilizes bidirectional attention to generate the next patch. To investigate the impact of patch size, we vary it while keeping the model's total parameter count constant. As illustrated in Figure 3, performance declines when patch sizes are either too large or too small. Excessively small patches diminish the model's bidirectional

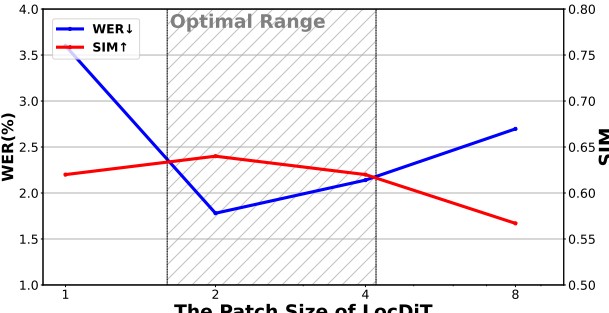

Figure 3: The impact of the patch size of LocDiT.

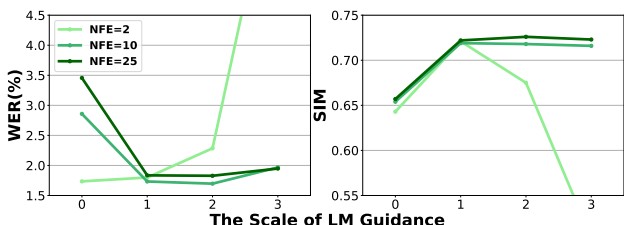

Figure 4: The impact of LM guidance under different NFE setups. $w = 0$ indicates that guidance is not used.

attention capability, forcing reliance on causal-attention AR and degrading performance. This may explain the poor performance of causal AR with diffusion loss as noted in Li et al. (2024a). Conversely, overly large patches turn LocDiT into a bottleneck, necessitating increased parameters.

**The Number of Historical Patches of LocDiT.** We experiment with different numbers of historical patches and confirm the critical role of LocDiT's context. Table 5 indicates that without historical context, the model's performance drastically worsens. Integrating historical context shifts LocDiT's function from mere generation to outpainting, markedly improving synthesis results. For larger patch sizes, such as 4, choosing a single historical context strikes the best balance between computational efficiency and performance.

**LM Guidance.** Figure 4 illustrates that LM guidance significantly enhances the diffusion decoder's inference process. Without guidance ($w = 0$), both WER and SIM deteriorate significantly. Conversely, an excessively large guidance scale can also impair outcomes. Additionally, even with extremely low NFE (such as 2), DiTAR still performs well on WER and SIM when combined with LM guidance.

### 4.4. Inference Analysis

#### 4.4.1. THE IMPACT OF TEMPERATURE

Temperature is vital for balancing diversity and determinism during language model inference, yet its definition for continuous-valued LMs remains underexplored. In this

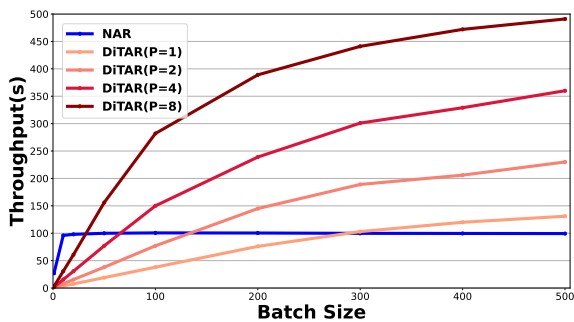

Figure 5: Throughput variations across different models as batch size changes.

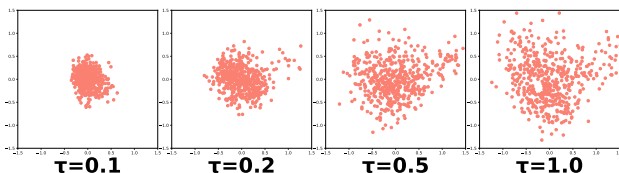

Figure 6: The impact of temperature on generation diversity.

study, we define temperature $\tau$ as the point at which random sampling is introduced during the reverse ODE solution.

To explore how $\tau$ balances diversity and stability, we synthesize the same text 500 times under different $\tau$ settings to assess voice diversity. We use only the text as DiTAR's prefix input, prompting the model to generate speech in random voices autoregressively. We then use WavLM-large to extract speaker embeddings from these samples. Following this, we conduct Principal Component Analysis (PCA), previously fitted on the training set, on the embeddings, and visualize the first two components. Figure 6 demonstrates that as $\tau$ increases, so does the diversity of the speaker voices.

We further test how the model's objective metrics change under various temperatures. As shown in Table 6, across different temperatures, the model consistently achieves favorable objective metrics on a large-scale test set. There is a trend that higher temperatures yield slightly better SIM scores, whereas lower temperatures result in better WER scores. The underlying reason may be that simulating the voice of unseen speakers requires greater diversity from the model, while pronunciation robustness demands more determinacy and stability from the model.

### 4.4.2. EFFICIENCY

We consider multiple inference metrics, including: 1) throughput: the duration of audio generated per unit time. 2) Real Time Factor (RTF): the ratio of the generation time to the audio length. 3) latency: the time required to output the first frame of audio (excluding the vocoder). We compare

Table 6: Objective evaluation results under different temperature values and NFE.

| $\tau$ | NFE=2 | | NFE=10 | |
|---|---|---|---|---|
| | **WER(%)↓** | **SIM↑** | **WER(%)↓** | **SIM↑** |
| 0 | 1.666 | 0.717 | 1.623 | 0.719 |
| 0.5 | 1.669 | 0.722 | 1.699 | 0.727 |
| 1 | 1.686 | 0.72 | 1.689 | 0.727 |

Table 7: Comparison of latency and RTF under batch sizes of 500 and 1.

| **System** | **Latentcy(s)↓** | **RTF↓** |
|---|---|---|
| Batch size = 500 | | |
| NAR | 50 | 5.03 |
| DiTAR (P=4) | 0.14 | **1.39** |
| DiTAR (P=2) | **0.11** | 2.17 |
| Batch size = 1 | | |
| NAR | 0.37 | **0.037** |
| DiTAR (P=4) | 0.066 | 0.66 |
| DiTAR (P=2) | **0.064** | 1.28 |

the inference performance of NAR (a diffusion transformer) and DiTAR under the same parameter sizes. During inference, all systems use CFG with NFE = 10. We evaluate all metrics by inferring 10 seconds of audio on an A100 GPU.

As shown in Figure 5, with small batch sizes, NAR achieves higher throughput due to lacking autoregressive computations. As batch sizes grow, NAR's high FLOPs demands hinder throughput gains, while DiTAR's throughput increases rapidly and is significantly superior to NAR.

DiTAR, blending a language model with a diffusion transformer, inherits features from both components. As shown in the Table 7, DiTAR always has lower latency than NAR due to its autoregressive nature. In terms of RTF, NAR has high parallelism and can achieve fast speed with a small batch size. In DiTAR, the degree of parallelism can be adjusted by changing the patch size, allowing for a trade-off between latency and RTF.

## 5. Conclusion

In this work, we propose DiTAR, a patch-based autoregressive framework combining a language model with a diffusion transformer. This approach significantly enhances the efficacy of autoregressive modeling for continuous tokens and reduces computational demands. For inference, we introduce temperature as the introduction time point for noise while solving the reverse diffusion ODE. Applied to zero-shot speech synthesis, DiTAR achieves SOTA robustness, speaker similarity, and naturalness with substantially lower computational requirements.

# Impact Statement

When DiTAR is applied to zero-shot voice synthesis, due to the realistic quality and high fidelity of the generated speech, there is a potential risk of its misuse for malicious purposes. When implementing in applications, it is crucial to strictly control the voices being cloned, such as through manual or automated review processes.

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

# A. Implementation Details

## A.1. Training Details

Regarding data processing for LibriLight, since it lacks transcriptions, we use our internal ASR system to transcribe this dataset and then convert the text into phonemes for use. For Emilia, we perform Grapheme-to-Phoneme (G2P) conversion on the official transcriptions provided.

We utilize 16 A100 GPUs, each processing a batch size of 15K tokens, and train DiTAR for 0.5M steps. The AdamW optimizer is employed with a constant learning rate of 1e-4, $\beta_t = 0.9$, and $\beta_2 = 0.99$. For DiTAR with 1B parameters, we utilize 32 A100 GPUs with a batch size of 7.5k per GPU.

## A.2. Model Configuration for Scaling

During the validation of DiTAR's scaling behavior, we trained models of four different sizes, ranging from 0.1 billion to 1 billion parameters. Specific hyperparameter configurations are detailed in Table 8.

Table 8: Configurations of DiTAR with different sizes.

| Model size | | $\sim 0.1$B | $\sim 0.4$B | $\sim 0.6$B | $\sim 1$B |
|---|---|---|---|---|---|
| Hyper-parameters | | Value | | | |
| Encoder | Number of layers | 4 | 4 | 6 | 8 |
| | Hidden dim | 512 | 1024 | 1024 | 1024 |
| | Number of heads | 8 | 16 | 16 | 16 |
| | FFN dim | 2048 | 4096 | 4096 | 4096 |
| Language Model | Number of layers | 24 | 24 | 36 | 24 |
| | Hidden dim | 512 | 1024 | 1024 | 1536 |
| | Number of heads | 8 | 16 | 16 | 24 |
| | FFN dim | 1024 | 4096 | 4096 | 6144 |
| LocDiT | Number of layers | 4 | 4 | 6 | 8 |
| | Hidden dim | 512 | 1024 | 1024 | 1024 |
| | Number of heads | 8 | 16 | 16 | 16 |
| | FFN dim | 2048 | 4096 | 4096 | 4096 |

## A.3. Derivation of $x_\theta$ in Temperature Sampling for Different Parameterized Diffusion

As previously discussed in Eq. 5, $x_\theta$ denotes the predicted data, which can be derived under different parameterizations of diffusion. Considering a diffusion process defined by $x_t = \alpha_t x_0 + \sigma_t \varepsilon$, where $x_0 \sim q(x_0)$ denotes the data, $\varepsilon \sim N(0, \mathbf{I})$ denotes the standard Gaussian noise, $t \in [0, 1]$.

For $\epsilon$-prediction mode,

$$x_\theta(x_t, t) = \frac{x_t - \sigma_t \epsilon_\theta(x_t, t)}{\alpha_t} \tag{8}$$

For $x_0$-prediction mode, $x_\theta(x_t, t)$ is exactly the model's prediction.

For $v$-prediction mode, also known as flow-matching. Next, we will proceed with a step-by-step deduction. Based on the definition of $v$, we can derive the following:

$$v = \dot{\alpha}_t x_0 + \dot{\sigma}_t \varepsilon \tag{9}$$

$$= \dot{\alpha}_t x_0 + \frac{\dot{\sigma}_t(x_t - \alpha_t x_0)}{\sigma_t} \tag{10}$$

$$= (\dot{\alpha}_t - \frac{\dot{\sigma}_t \alpha_t}{\sigma_t})x_0 + \frac{\dot{\sigma}_t}{\sigma_t}x_t \tag{11}$$

After rearanging $v$ and $x_0$, we get:

$$x_0 = \frac{\dot{\sigma}_t x_t - \sigma_t v}{\sigma_t \dot{\alpha}_t - \dot{\sigma}_t \alpha_t} \tag{12}$$

Thus,

$$x_\theta(x_t, t) = \frac{\dot{\sigma}_t x_t - \sigma_t v_\theta(x_t, t)}{\sigma_t \dot{\alpha}_t - \dot{\sigma}_t \alpha_t} \tag{13}$$

## A.4. Calculation of FLOPs

When calculating FLOPs for all systems, we use a 3-second audio prompt to synthesize 10 seconds of audio. Assuming a text frame rate of 7Hz, the prompt's text length is 21, and the target text length is 70. We focus solely on the computationally dominant components, including causal transformers, non-causal transformers, and convolutional layers. For causal transformers, calculations account for the use of KV cache. The multi-head setup does not impact FLOPs, so we uniformly assume that the number of heads is 1. Considering that bias and normalization layers contribute minimally to the overall FLOPs of the transformer, we will temporarily disregard them. Matrix multiplication involves an equal number of additions and multiplications, so the result should be multiplied by 2. For the diffusion part, the corresponding FLOPs need to be multiplied by the Number of Function Evaluations (NFE). Additionally, Classifier-Free Guidance (CFG) incurs double the computational cost.

For a one-dimensional convolution network with $N$ layers, a hidden channel size of $C$, a kernel size of $K$, a stride size of 1, and the input length of T, the FLOPs can be calculated as follows:

$$\text{FLOPs} = [C, KT] \times [KT, C] \times N = 2C^2 KTN \tag{14}$$

For a non-causal transformer with $N$ layers, a hidden size of $C$, an FFN intermediate hidden size of $C_{\text{mid}}$, and the input length of T, the FLOPs can be calculated as follows:

$$\text{QKV} = [T, C] \times [C, 3C] = [T, 3C] = 6TC^2 \tag{15}$$

$$\text{Attention} = QK^T = [T, C] \times [C, T] = [T, T] = 2T^2 C \tag{16}$$

$$\text{V} = [T, T] \times [T, C] = [T, C] = 2T^2 C \tag{17}$$

$$\text{FC} = [T, C] \times [C, C] = [T, C] = 2TC^2 \tag{18}$$

$$\text{FFN\_FC1} = [T, C] \times [C, C_{\text{mid}}] = [T, C_{\text{mid}}] = 2TCC_{\text{mid}} \tag{19}$$

$$\text{FFN\_FC2} = [T, C_{\text{mid}}] \times [C_{\text{mid}}, C] = [T, C] = 2TCC_{\text{mid}} \tag{20}$$

$$\text{Transformer\_FLOPs}(N, C, T, C_{\text{mid}}) = (\text{QKV} + \text{Attention} + \text{V} + \text{FC} + \text{FFN\_FC1} + \text{FFN\_FC2}) \times N \tag{21}$$

$$\tag{22}$$

For a causal transformer with $N$ layers, a hidden size of $C$, an FFN intermediate hidden size of $C_{\text{mid}}$, a prefix input length of $T_{\text{pre}}$, and the input length of T, the FLOPs can be calculated as follows:

$$\text{Prefix\_FLOPs} = \text{Transformer\_FLOPs}(N, C, T_{\text{pre}}, C_{\text{mid}}) \times N \tag{23}$$

$$\text{QKV} = [1, C] \times [C, 3C] = [1, 3C] = 6C^2 \tag{24}$$

$$\text{Attention} = QK^T = [1, C] \times [C, t] = [1, t] = 2t^2 C \tag{25}$$

$$\text{V} = [1, t] \times [t, C] = [1, C] = 2t^2 C \tag{26}$$

$$\text{FC} = [1, C] \times [C, C] = [1, C] = 2C^2 \tag{27}$$

$$\text{FFN\_FC1} = [1, C] \times [C, C_{\text{mid}}] = [1, C_{\text{mid}}] = 2CC_{\text{mid}} \tag{28}$$

$$\text{FFN\_FC2} = [1, C_{\text{mid}}] \times [C_{\text{mid}}, C] = [1, C] = 2CC_{\text{mid}} \tag{29}$$

$$\text{AR\_FLOPs}(N, C, T, T_{\text{pre}}, C_{\text{mid}}) = \sum_{t=1+T_{\text{pre}}}^{T+T_{\text{pre}}} (\text{QKV} + \text{Attention} + \text{V} + \text{FC} + \text{FFN\_FC1} + \text{FFN\_FC2}) \times N \tag{30}$$

$$\text{AR\_Transformer\_FLOPs}(N, C, T, T_{\text{pre}}, C_{\text{mid}}) = \text{Prefix\_FLOPs} + \text{AR\_FLOPs} \tag{31}$$

# B. Subjective Evaluation

Apart from objective metrics, we employ many subjective metrics to comprehensively evaluate the generation ability of different systems. In the process of evaluation, raters are presented with 2 audios for comparison and 1 audio as a reference. Each rater is asked to score after comparing the two audios (CMOS) and to also rate each audio individually (N-MOS, Q-MOS, S-MOS). The user interface is shown in Figure 7 and 8.

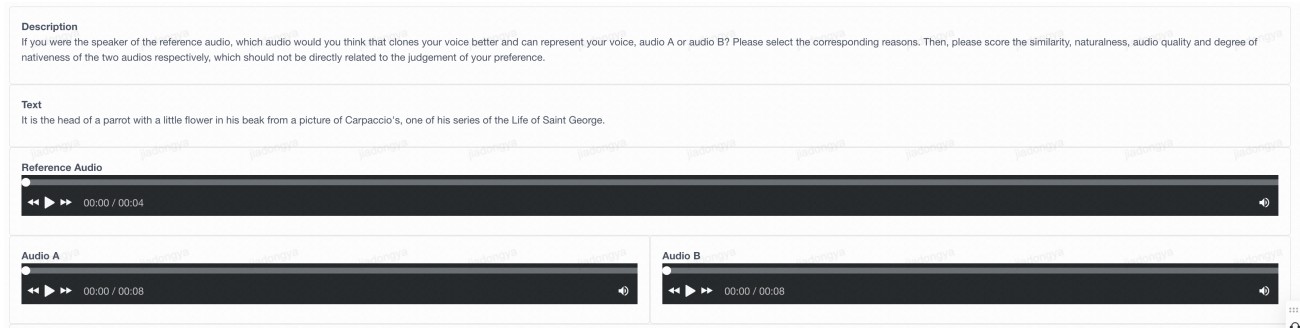

Figure 7: The user interface for subjective evaluation.

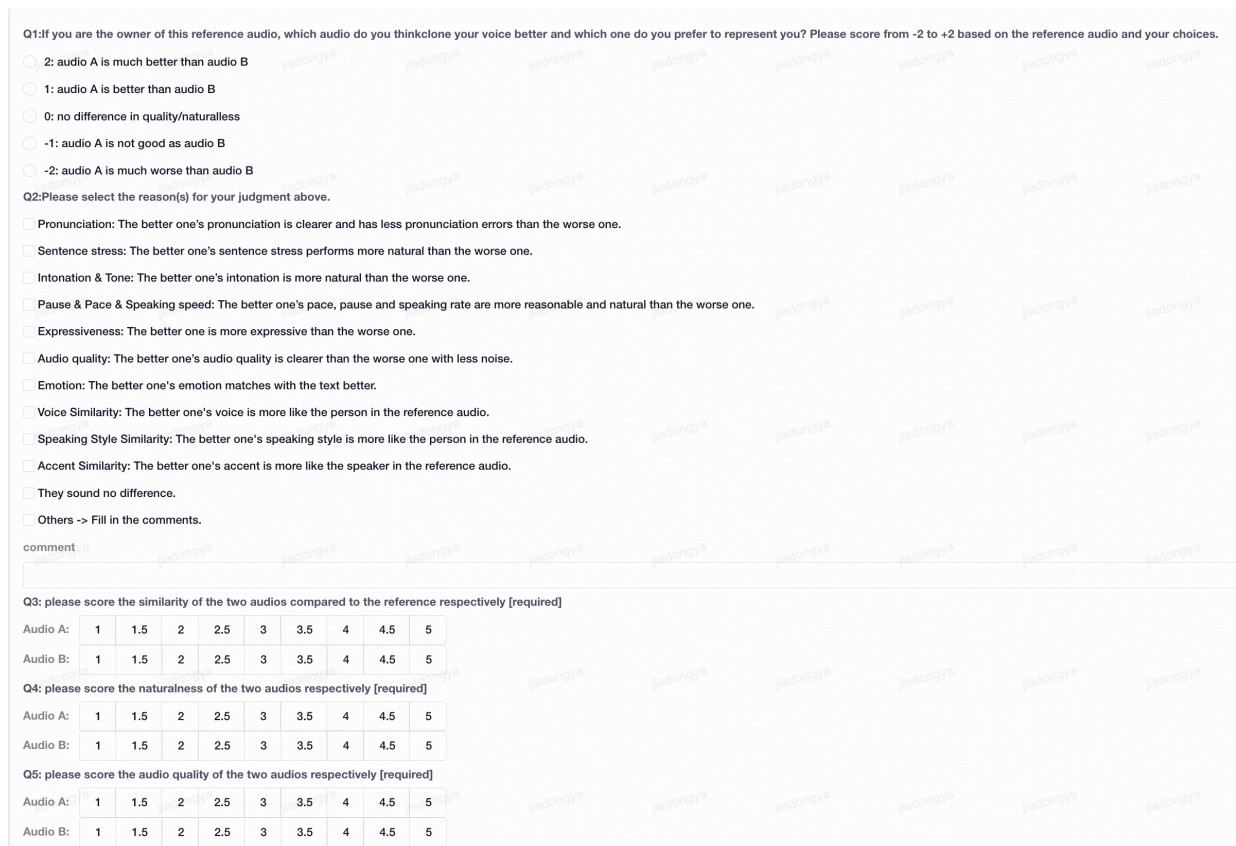

Figure 8: The rating interface and questions presented to raters.

## C. Comparison with Other Autoregressive Diffusion Works

Different systems have varying design philosophies. MAR and DiTAR offload the computation of diffusion to the diffusion head, while ARDiT applies diffusion throughout the entire model. DiTAR structurally resembles a causal language model and becomes a continuous-valued LLM when scaled.

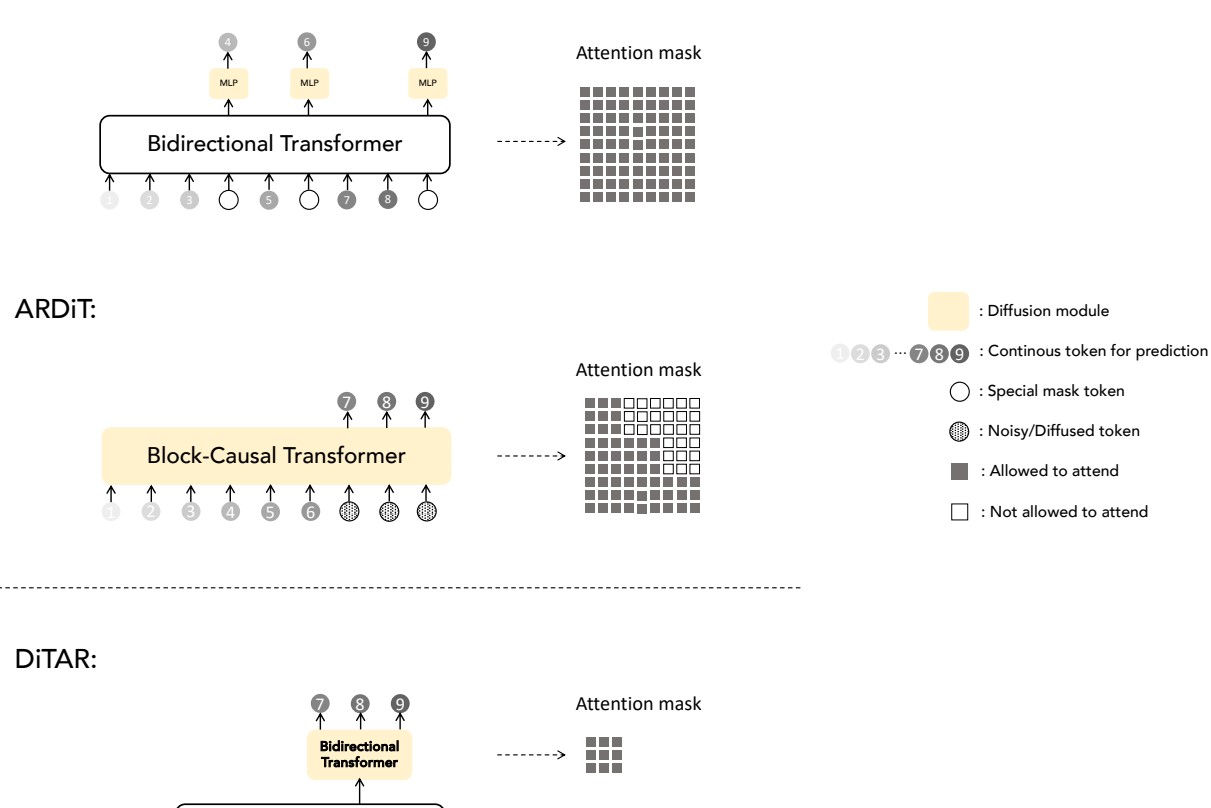

Figure 9: Comparison of different frameworks.

