# OpenReview forum: "DiTAR: Diffusion Transformer Autoregressive Modeling for Speech Generation"
_ICML.cc/2025/Conference — ICML 2025 poster_

### Official Review · Reviewer_UCo6 · 2025-02-14

**Overall Recommendation:** 4

**Summary:**

The paper presents DiTAR (Diffusion Transformer Autoregressive Modeling), a novel approach that combines an autoregressive language model (LM) with a diffusion transformer (LocDiT) to improve continuous speech generation. The key idea is a patch-based modeling strategy, where the LM predicts the sequence at a high level, and LocDiT refines the details within each patch using bidirectional attention.

A temperature-based sampling method is also introduced to control the trade-off between determinism and diversity in the generation process. Evaluations on zero-shot TTS benchmarks demonstrate that DiTAR achieves state-of-the-art performance in robustness, speaker similarity, and naturalness, while requiring significantly less computation than competing methods like Voicebox and NaturalSpeech.

**Claims And Evidence:**

The paper makes several claims, most of which are supported by strong empirical evidence. The claim that DiTAR outperforms existing zero-shot TTS models is well-supported by both objective and subjective evaluations. Word Error Rate (WER), speaker similarity, and UTMOS scores confirm that DiTAR produces more robust and natural speech compared to previous baselines. The claim that DiTAR reduces computational costs is backed by FLOPS measurements and throughput comparisons, showing that it achieves similar or better performance with up to 43× lower compute requirements than non-autoregressive diffusion models.

Another claim, that temperature-based sampling is essential for balancing diversity and determinism in continuous-valued LMs, is supported by a PCA analysis of generated speaker embeddings, showing that different temperatures influence the diversity of generated voices. The modified Classifier-Free Guidance (CFG) method is another strong contribution, making it more suitable for patch-based diffusion models.

However, certain aspects of the methodology could be explained more clearly. Some theoretical transitions feel abrupt, such as the statement that “operating in the velocity space with a conditional flow-matching target is also equivalent”, which lacks context or formal justification. The role of historical patches, denoted as  h_{i-2}, h_{i-1}  in Figure 1, is also not fully formalized in Section 3.1. Readers can infer how they are encoded and used, but a clearer explanation in the main text would improve clarity.

**Essential References Not Discussed:**

N/A

**Experimental Designs Or Analyses:**

The experimental setup is thorough and provides compelling evidence for the effectiveness of DiTAR. Evaluations are conducted across multiple zero-shot TTS benchmarks, ensuring a fair comparison with strong baselines. The ablation studies on patch size, historical context, and LM guidance are insightful and highlight key trade-offs in model design.

The scaling analysis further strengthens the paper’s claims, showing that WER and speaker similarity improve consistently as the model size and training data increase. A discussion on whether performance would continue improving with larger models (e.g., 10B+ parameters) would be valuable.

**Methods And Evaluation Criteria:**

The evaluation of DiTAR is comprehensive and includes multiple zero-shot TTS benchmarks, comparing its performance against strong baselines like Voicebox and NaturalSpeech3. The results demonstrate that DiTAR achieves state-of-the-art performance in robustness, speaker similarity, and naturalness while maintaining significantly lower computational costs. The analysis of patch size versus historical context is particularly insightful, providing valuable guidance on how to balance computational efficiency with generation quality.

**Other Comments Or Suggestions:**

I am not an expert in ODE/SDE solver for diffusion model (though I am fairly familiar with DDPM, DDIM, and major Flow-matching objectives). I am not sure if the temperature-based sampling introduced in this paper is already present somewhere in the literature.

**Other Strengths And Weaknesses:**

This paper's architecture can be used for Speech Language model as well to speed up the inference and improve speech synthesis quality.

**Questions For Authors:**

- Do you have ablation experiments on aggregation encoder to make it causal? One claim made by the authors is that causal attention degrades performance in diffusion-based autoregressive modeling, so it will be nice to have some comparisons here.
- I am curious what happens if the model (and its training data) is scaled up further? (since currently it’s only scaled until 1B).

**Relation To Broader Scientific Literature:**

This work is highly relevant to recent advances in diffusion-based generative models and speech synthesis. It builds upon prior work like ARDiT, NaturalSpeech3, and Transfusion, but improves efficiency by combining diffusion with autoregressive patch modeling. The use of patchification is inspired by techniques in image and video generation, such as those found in latent diffusion models, and applies them effectively to speech synthesis.

The paper also contributes to research on temperature-based sampling in diffusion models. While the idea of controlling randomness through temperature is well-known in SDE-based diffusion models, the specific approach of defining temperature as a noise injection point in ODE solvers is a novel adaptation.

**Theoretical Claims:**

N/A (there is no theoretic proof, the paper is mostly empirical). I checked the derivation of the proposed temperature-based sampling process and it looks good to me.

---

> ### Author Rebuttal · Authors · 2025-04-01
>
> We sincerely appreciate your positive review and insightful comments.
> Most of your points are aligned with the contributions we aim to convey in our paper. Next, we address your questions organized according to the review sections.
>
> We have attached audio samples of our method in this link:https://spicyresearch.github.io/ditar/#hard-cases. Feel free to listen.
>
> **Questions in "Claims And Evidence"**
>
> Operating velocity in classifier-free guidance (CFG)：
> In the paper, we derive the CFG process based on the score to align with the original work [1] for easier understanding. The score can be easily converted to velocity, allowing us to straightforwardly apply the CFG method to flow-matching or v-prediction models. Below, we provide the derivation process, which will be added to the paper later.
>
> Begin with the definition of velocity $v$:
>
> $ v= \\dot{\\alpha_t}x_0 + \\dot{\\sigma_t}\\epsilon=\\dot{\\alpha_t}\\frac{x_t-\\sigma_t\\epsilon}{\\alpha_t}+\\dot{\\sigma_t}\\epsilon$
>
> Rearrange concerning $\epsilon$：
>
> $\\epsilon = \\frac{\\alpha_t v-\\dot{\\alpha}_t x_t}{\\alpha_t\\dot{\\sigma}_t - \\dot{\\alpha_t}\\sigma_t}$
>
> Perform CFG on score space and substitute the above equation：
>
> $ \\tilde{\\epsilon}( x_t, c )  = (1+w) \\epsilon( x_t,c ) - w\\epsilon(x_t) = \\frac{\\alpha_t \\tilde{v} ( x_t, c) - \\dot{\\alpha}_t x_t}{\\alpha_t\\dot{\\sigma}_t - \dot{\\alpha_t}\\sigma_t}$
>
> where
>
> $ \\tilde{v}( x_t, c ) = ( 1+w )v(x_t, c) - w v( x_t)$
>
>
> Therefore, performing CFG operations in the velocity space is equivalent to doing so in the score space.
>
> **Questions in "Questions For Authors"**
> -   Validate the claim that causal attention degrades the performance of continuous-valued AR:
>     -   We experimentally found that the aggregation encoder has a minimal impact on the receptive field. Given that LocDiT uses historical patches and non-causal attention, even if the aggregation encoder is a causal transformer, the impact is minor.
>     - The aggregation encoder is not a primary innovation of this work, and as shown in Table 3, the benefits from scaling the encoder are small.
>     -   We have validated this claim from another perspective. In Table 4 of the paper, as the patch size decreases and the number of historical patches reduces, the entire model becomes more causal. When the patch size is set to 1 and the number of historical patches is set to 0, the model turns into a vanilla causal language model. It can be observed that the more causal the model, the worse the performance.
> -   Further scaling：
>     -   In the zero-shot TTS task, the amount of training data is limited and the task is relatively well-defined. Further scaling of the model provides marginal benefits, considering inference performance. Therefore, we did not pursue scaling beyond 1B parameters.
>     -   Our framework is a general generative model, not limited to speech generation. We aim to apply it to more complex tasks, such as speech LLM and video generation. As a continuous-valued LM, we hope it will achieve scaling performance comparable to discrete-valued LMs.
>
> **References**
>
> [1] Ho, Jonathan, and Tim Salimans. "Classifier-free diffusion guidance." _arXiv preprint arXiv:2207.12598_ (2022).
>
> *We sincerely hope that our reply could address your concerns and that you might consider raising the rating. Please let us know if you have any further questions or require additional results.*

---

### Official Review · Reviewer_Pa2m · 2025-03-13

**Overall Recommendation:** 3

**Summary:**

This paper proposes DiTAR (Diffusion Transformer Autoregressive Modeling), a patch-based autoregressive framework for zero-shot text-to-speech synthesis that combines language models with diffusion transformers. The method uses a divide-and-conquer strategy where continuous speech tokens are partitioned into patches. A language model handles inter-patch prediction, while a localized diffusion transformer (LocDiT) with bidirectional attention generates each patch. The authors introduce a temperature-based sampling approach for the continuous-valued autoregressive model and demonstrate superior scaling properties. According to the authors' evaluation, DiTAR achieves state-of-the-art performance in zero-shot speech generation for robustness (WER), speaker similarity (SIM), and naturalness with reduced computational demands compared to existing models.

## update after rebuttal

The authors' responses have addressed most of my concerns. I have raised my score from 2 to 3.

**Claims And Evidence:**

Several key claims in the paper lack sufficient supporting evidence or comparative analysis:

1. The claim of novelty in the patch-based approach is weakened by insufficient comparison to similar prior work, particularly VALL-E 2's Grouped Code Modeling (Chen et. al 2024b), which implements a comparable approach. As shown in the VALL-E 2 paper: "We partition the codec code sequence into groups with the group size G, and C0:G stands for the group [c0, c1, ..., c(G-1)]." This approach appears functionally equivalent to DiTAR's next-patch-prediction method, yet this similarity is not acknowledged.

2. The computational efficiency claims are supported by FLOPS calculations but are problematic as they only materialize at unrealistically large batch sizes (>100) for optimal patch size (2 or 4) that would exceed typical GPU memory constraints in production environments (unless using GPUs such as H200 or B200). Figure 5 clearly shows that for batch sizes below 100, NAR models maintain superior or very close throughput.

3. There are unexplained inconsistencies in the reported results between tables (e.g., the 0.4B model shows WER of 1.876 in Table 3 but 1.685 in Table 6, and SIM of 0.716 vs. 0.735, which I assume both are on SeedEval dataset), undermining confidence in the reliability of the findings.

4. The subjective evaluation claims (naturalness, quality) cannot be independently verified due to the absence of audio samples (demo page), which is a significant limitation for a text-to-speech paper.

**Essential References Not Discussed:**

The paper inadequately discusses or compares to several highly relevant references:

1. VALL-E 2 (Chen et al., 2024) introduces Grouped Code Modeling, which is remarkably similar to DiTAR's patch-based approach. Despite being trained on the same LibriLight dataset and achieving comparable performance (WER of 1.5 and SIM of 0.64), this paper is not sufficiently compared against it.

2. ARDiT's approach of diffusion-based autoregressive generation deserved more direct comparison, particularly regarding whether the encoding step in DiTAR provides meaningful benefits over ARDiT's more straightforward approach.

**Experimental Designs Or Analyses:**

Several issues affect the experimental validity:

1. The comparison with competing methods is incomplete, with notable omissions such as VALL-E 2, which reports similar performance metrics (WER of 1.5 and SIM of 0.64) with slightly better WER trained on the same LibriLight dataset that DiTAR uses.

2. The computational efficiency analysis is conducted under conditions that favor the method but are impractical for real-world deployment. Specifically, the throughput comparison in Figure 5 demonstrates that DiTAR with patch sizes of 2 or 4 only becomes more efficient than NAR models at batch sizes exceeding 100. This requirement is unrealistic for several reasons:

   - Memory constraints: Most GPUs have at most approximately 80GB RAM per card, which is insufficient for batch sizes of 100+ when accounting for model parameters, activations, and KV cache, especially for long-form speech generation.

   - Distributed inference overhead: Linking multiple GPUs for distributed inference introduces significant communication overhead, which is not factored into the throughput calculations.

   - Practical deployment considerations: In production environments, it's typically more efficient to distribute smaller batches across multiple independent GPUs than to process large batches with linked GPUs due to reduced latency and better resource utilization.

   - The efficiency claims would only be realized on specialized high-end hardware like NVIDIA B200 or H200 GPUs, which represents an impractical deployment target for most applications.


3. The inconsistency in reported performance metrics between different tables (Tables 1, 3, and 6 all report different WER and SIM) raises questions about the reliability of the results.

**Methods And Evaluation Criteria:**

The methods are generally sound, although some evaluation aspects are questionable:

1. The benchmark datasets and metrics (WER, SIM, UTMOS) are appropriate for TTS evaluation.

2. However, the throughput/efficiency metric (FLOPS) is implemented in a way that favors the proposed approach under unrealistic conditions (very large batch sizes). Specifically:

   - As shown in Figure 5, DiTAR only surpasses NAR models in throughput at high batch sizes (around 100) for optimal patch size ranges (2 or 4), which is impractical for most deployment scenarios.

   - Most production-grade GPUs (like A100 with 80GB memory) cannot accommodate such large batch sizes for these models, especially for long-form speech generation where the KV cache assumption in FLOPS calculation is useful.

   - When accounting for model parameters, optimizer states, and gradient accumulation, batch sizes of 100+ would require multiple high-end GPUs operating in parallel, introducing communication overhead that negates the theoretical throughput advantages.

   - In real-world deployment scenarios, lower latency with smaller batch sizes is often preferred to higher throughput with large batches as it can be distributed across multiple low VRAM GPUs for inference, making NAR models more practical despite their theoretical inefficiency with large batch sizes.

3. The lack of detailed ablation studies comparing the full architecture (encoding → AR → diffusion) to simpler alternatives (like direct AR → diffusion as in ARDiT) prevents a clear understanding of whether the added complexity is necessary, especially given LocDiT needs historical patches as conditions for diffusion, making the entire framework more similar to ARDiT (Liu et. al, 2024b) than Li et. al, 2024a.

**Other Comments Or Suggestions:**

- Since LocDiT also relies on historical patches, the statement "while $\theta_b$ denotes a bidirectional-attention diffusion transformer executing next patch prediction via \$p_{\theta\_b}(\mathbf{x}\_{i+1}, \ldots, \mathbf{x}\_{i+P} | \mathbf{h}_i)\$" should be replaced with

\$p_{\theta\_b}(\mathbf{x}\_{i+1}, \ldots, \mathbf{x}\_{i+P} | \mathbf{h}_i, \mathbf{x}\_{i}, \ldots, \mathbf{x}\_{i-K})\$ where $K$ is your historical patch size.

- Eq. 7 does not seem to be particularly "flow-matching (Lipman et al., 2022)" but more like velocity prediction (Salimans et al. 2022)

**Other Strengths And Weaknesses:**

Strengths:
1. The paper presents a coherent framework integrating language models and diffusion models.

2. The temperature-based sampling approach for continuous-valued autoregressive models is an interesting contribution. However, it seems to be contrived for the diversity purpose since $\tau = 1$ corresponds to the original DDIM sampling algorithm (such as Cosyvoice or Seed-TTS), so the baseline it compares to ($\tau = 0$) is artificially limited in diversity.

3. The scaling analysis is comprehensive and demonstrates good scaling properties.

Weaknesses:

1. Novelty is not well justified, with insufficient acknowledgment of similar prior approaches such as Vall-E 2 and ARDiT.

2. The computational efficiency claims are presented in a way that overstates practical benefits. The throughput advantage only materializes at batch sizes exceeding 100, which is impractical for most deployment scenarios due to memory constraints (80GB per GPU) and the inefficiency of distributed inference for such tasks.

3. Inconsistencies in reported metrics undermine confidence in the results. The significant variations between Table 3 and Table 6 for the same 0.4B model (WER: 1.876 vs 1.685; SIM: 0.716 vs 0.735) cannot be explained by random variation, and Table 6 is the only one that compares to more AR systems other than outdated models such as Vall-E.

4. The absence of audio samples (demo page) prevents verification of subjective quality claims, which is particularly problematic for a text-to-speech paper where perceptual quality is paramount.

5. The necessity of the three-step architecture versus simpler alternatives (ARDiT) is inadequately justified. The paper does not explore whether the encoding → AR → diffusion approach provides meaningful advantages over simpler approaches like direct AR → diffusion used in ARDiT.

**Questions For Authors:**

1. What is the use of the encoded patch information if we still need historical patches (non-encoded patches) when we call the diffusion model? How does it differ from ARDiT and how is it better than ARDiT?

2. Could you explain the discrepancies in reported metrics between tables (e.g., the 0.4B model's WER and SIM metrics in Tables 3 vs. 6)?

3. Can you make audio samples available to verify the subjective quality claims?

3. What are $\dot{\alpha_t}$ and $\dot{\sigma_t}$ at line 269-270, page 5, right column?

**Relation To Broader Scientific Literature:**

The paper builds upon two major approaches in speech synthesis: autoregressive language models and diffusion models. While it cites many relevant papers, it insufficiently contextualizes its contribution relative to recent advances that use similar techniques:

1. The patch-based AR approach bears strong similarity to VALL-E 2's Grouped Code Modeling, which similarly divides codec codes into grouped patches processed sequentially.

2. The use of diffusion for patch prediction resembles existing approaches like ARDiT and Transfusion, but the paper does not sufficiently explore whether their three-stage approach (encoding → AR → diffusion) provides meaningful advantages over the simpler AR → diffusion methodology in ARDiT.

**Theoretical Claims:**

The paper does not contain theoretical claims.

---

> ### Author Rebuttal · Authors · 2025-03-31
>
> Thank you for your insightful comments. We provide detailed responses to your concerns as summarized below:
>
> **Q1. Subjective evaluation**
>
> Audio samples can be found in this link: https://spicyresearch.github.io/ditar/#hard-cases
>
> **Q2. Connection with VALL-E 2**
>
> They are different in many aspects.
> - VALL-E	2 is a two-stage (AR+NAR) method for discrete tokens, whereas DiTAR is a single-stage (AR) method for continuous tokens.
> -  Patchification serves different purposes in the two methods. For VALL-E 2, it reduces computational load, whereas for DiTAR, it enables bidirectional modeling for next-patch prediction and overcomes the limitations of causal LMs.
>
> **Q3. Connection with ARDiT**
>
> Although both DiTAR and ARDiT have autoregressive and diffusion elements, they have completely different design philosophies.
> - The core difference lies in _which part of the model acts as the diffusion component_. The figure in the link better illustrates their differences: https://spicyresearch.github.io/ditar/#comparison
> - ARDiT is a diffusion model throughout its entire architecture.
> - Differently, DiTAR is essentially a language model with a diffusion head. The computational load of multi-step sampling in diffusion has been shifted to the diffusion head.
>
> **Q4. Discussion of computational efficiency**
>
> Thank you for your detailed response. Some of your points are insightful but contradict our actual experimental results, we elaborate in detail below:
> - Memory constraints: All our tests were conducted on a standard A100 GPU (80G Memory), with batch sizes ranging from 0 to 500, and no out-of-CUDA-memory incidents occurred. The model we used is a 400M parameters transformer, which is a common size in zero-shot TTS task [2][3]. Theoretically, a large batch size is practically reasonable for this size of model.
> - Practical deployment considerations: There is no need for specialized high-end hardware like H200. All our tests were conducted on a standard A100 GPU with 80G.
> - Distributed inference overhead: For commonly used TTS models like a 400M parameters transformer, this size typically doesn't require distributed inference.
> - Latency considerations: Different from NAR, DiTAR can maintain very low latency even with large batch sizes. (please see Q4 in the response to review oFFs).
> - We do not intend to prove that DiTAR is superior to NAR diffusion under all levels of concurrency. The insight we want to convey is that DiTAR is a model positioned between NAR and AR: it has low latency and high throughput like AR, while increasing parallelism by enlarging the patch size.
>
> **Q5. About the misunderstanding of inconsistency in different tables**
>
> Thank you for noticing the details. To clarify, different tables serve different purposes, which is why we have used different setups for each.
> - Table1:
> 	- Purpose: maximize fairness and align with other systems.
> 	- Setup: 0.6B; trained on Librilight/Emilia; evaluated on Librispeech test-clean subset A/B
> - Table3:
> 	- Purpose: Access the parameter scaling effects of different modules in DiTAR, so we start with a relatively smaller model and use a more difficult test set for evaluation.
> 	- Setup: 0.4B; trained on 280k hour data; evaluated on Seed
> - Table6:
> 	- Purpose: Assess the upper-bound performance of DiTAR by comparing DiTAR against various commercial proprietary models trained on various internal data.
> 	- Setup: 1B; trained on 280k hour data; evaluated on Seed
>
> **Q6. Objective comparison with VALL-E 2 and ARDiT**
>
> - VALL-E 2 have not released the checkpoints and the subset of the test set, so the scores reported in their papers cannot be directly used for comparison.
> - ARDiT is tested on a released subset of LibriTTS test-clean. We reevaluated the samples using the same tool.
>
> |Method|WER↓| SIM↑|
> |--|--|--|
> |ARDiT|4.036|0.613|
> |DiTAR(Ours)|**3.401**|**0.717**|
>
> **Q7. Response to other questions**
>
> - v-prediction v.s. flow matching: Under the same diffusion formulation defined by $a_t$ and $\sigma_t$, the v-prediction and flow-matching loss are mathematically equivalent [4]. We will provide the corresponding derivations in the paper.
> - $\dot{a}_t$ and $\dot{\sigma}_t$ are the first derivative of $a_t$ and $\sigma_t$ w.r.t $t$, respectively.
>
> **References**
>
> [1] Chen, Sanyuan, et al. "VALL-E 2: Neural codec language models are human parity zero-shot text to speech synthesizers." _arXiv preprint arXiv:2406.05370_ (2024).
>
> [2] Eskimez, Sefik Emre, et al. "E2 tts: Embarrassingly easy fully non-autoregressive zero-shot tts." _2024 IEEE SLT_. IEEE, 2024.
>
> [3] Chen, Yushen, et al. "F5-tts: A fairytaler that fakes fluent and faithful speech with flow matching." _arXiv preprint arXiv:2410.06885_ (2024).
>
> [4] Fu. Wang, et al. Rectified Diffusion: Straightness Is Not Your Need, ICLR 2025
>
> *We sincerely hope that our reply could address your concerns and that you might consider raising the rating. Please let us know if you have any further questions or require additional results.*

---

> > ### Comment · Reviewer_Pa2m · 2025-04-03
> >
> > I appreciate the authors' careful responses and thank them for their efforts in addressing my concerns. Here are my responses to the authors' rebuttal:
> >
> > 1. I appreciate the new demo page, which has partially addressed my concerns over the subjective evaluations. However, I noticed that the F5 and E2 have the same total duration while DiTAR does not. Since both F5 and E2 require a total duration input and current models do not support a duration predictor (while DiTAR has an internal "total duration predictor" since it can always sample a <EOS> token to get a total duration), is the comparison a little unfair? I think in the F5-TTS paper, the authors used the total duration of the ground truth. Could you please also generate some samples using the ground truth duration (or the same total duration as your samples)?
> >
> > 2. I believe the outcomes are quite similar since they both propose patched generation, even though the motivation and architecture are different.
> >
> > 3. I understand that DiTAR and ARDiT are different in ways the authors explain in the figure, but the authors did not address my main concern regarding the similarity between DiTAR and ARDiT. That is:
> >
> > > The use of diffusion for patch prediction resembles existing approaches like ARDiT and Transfusion, but the paper does not sufficiently explore whether their three-stage approach (encoding → AR → diffusion) provides meaningful advantages over the simpler AR → diffusion methodology in ARDiT.
> >
> > > The necessity of the three-step architecture versus simpler alternatives (ARDiT) is inadequately justified. The paper does not explore whether the encoding → AR → diffusion approach provides meaningful advantages over simpler approaches like direct AR → diffusion used in ARDiT.
> >
> > That is, I'm not concerned over its similarity to ARDiT but rather whether the newly proposed DiTAR is necessary compared to ARDiT.
> >
> > 4. I appreciate the authors' response regarding the throughputs, and it has addressed my concerns. I believe the authors should revise the paper to make this point clearer, especially by adding the discussion regarding the batch size and ARDiT's advantages/disadvantages over NAR, since the current version sounds more like an overstatement of the ARDiT's efficiency by ignoring its inefficiency in small batch sizes (for inferences in on-device situations, for example).
> >
> > 5. Thank you for your clarification. Could you please make the experimental setup clearer in your revised manuscript?
> >
> > 6.
> > > VALL-E 2 have not released the checkpoints and the subset of the test set, so the scores reported in their papers cannot be directly used for comparison.
> >
> > I believe the Vall-E 2 paper mentioned the evaluation models and test subset:
> >
> > > SIM is used to evaluate the speaker similarity between the original prompt and synthesized speech, leveraging the SOTA speaker verification model, WavLM-TDNN^3 [Chen et al., 2022]. The similarity score predicted by WavLM-TDNN is in the range of [−1, 1], with a larger value indicating higher speaker similarity.
> >
> > > WER (Word Error Rate) is used to evaluate the robustness of synthesized speech. Neural TTS systems sometimes experience deletion, insertion, and replacement errors due to incorrect attention alignments, which can affect their robustness. We perform ASR on the generated audio and calculate the WER with respect to the original transcriptions. In this experiment, we employ the open-sourced Conformer-Transducer model^4 [Gulati et al., 2020] as the ASR model.
> >
> > > Following Borsos et al. [2022] and Wang et al. [2023a], we use samples from LibriSpeech test-clean with lengths between 4 and 10 seconds, resulting in a 2.2 hours subset and 40 unique speakers. We evaluate each sample synthesis under two settings: 3s Prefix as Prompt and Ref Utterance as Prompt. For the first setting, we perform speech continuation and utilize the 3-second prefix of the speech as the prompt. In the second setting, we use a reference utterance from the same speaker as the prompt. Specifically, we begin by filtering the official speech list of LibriSpeech test-clean based on length. For the ordered speech list of each speaker, in the first setting, we synthesize the i-th speech sample using the first 3 seconds of the ground-truth i-th speech sample as the prompt. In the second setting, we synthesize the i-th speech sample using the (i − 1)-th sample as the prompt and synthesize the first speech sample using the last sample as the prompt.
> >
> > > 3: We use the best speaker verification model released at https://github.com/microsoft/UniSpeech/tree/main/downstreams/speaker_verification#pre-trained-models
> >
> > > 4: https://huggingface.co/nvidia/stt_en_conformer_transducer_xlarge
> >
> > Please check page 10 on https://arxiv.org/pdf/2406.05370 for more details.
> >
> > > ARDiT is tested on a released subset of LibriTTS test-clean. We reevaluated the samples using the same tool.
> >
> > ARDiT was trained on LibriTTS, while DiTAR was trained on LibriLight, making this comparison unfair.

---

> > > ### Author Response · Authors · 2025-04-04
> > >
> > > We appreciate your detailed response. Regarding your questions, our replies are as follows:
> > >
> > > 1. Subjective comparison with E2TTS and F5TTS:
> > > - To clarify, we are comparing the end-to-end text-to-speech performance between systems, and duration modeling is a part of the system. AR models can naturally simulate duration, while NAR models require an additional duration prediction module. If ground-truth (GT) duration is used for NAR systems, then for texts without GT audio, the duration cannot be obtained.
> > > - The duration of E2TTS and F5TTS: They share the same duration prediction method, based on F5TTS's released code and checkpoint. To clarify, the F5TTS paper uses rule-based predicted duration instead of GT duration, as mentioned on Page 4:
> > > > The sequence length N, or duration, has now become a pivotal factor that necessitates informing the model of the desired length for sample generation. One could train a separate model to predict and deliver the duration based on xr⁢e⁢f, yr⁢e⁢f and yg⁢e⁢n. Here we simply estimate the duration based on the ratio of the number of characters in yg⁢e⁢n and yr⁢e⁢f.
> > >  - We synthesized F5TTS and E2TTS samples using GT duration and have provided them at the following link：https://spicyresearch.github.io/temp_samples/
> > >
> > > 2. The impact of pachification is different.
> > > - VALL-E 2 achieves the best results with patch=1( Table1 in VALL-E 2 paper). The purpose of patchification is only to reduce computational load.
> > > - For DiTAR, the best results are achieved with patch>1. This demonstrates that DiTAR's patchification, which introduces bidirectional attention modeling within patches, improves performance.
> > > - Table 2 below also demonstrates this conclusion.
> > >
> > > 3. The advantage over ARDiT:
> > >
> > > Thank you for your perspective, this is a topic worth discussing.
> > > - We mentioned in the paper as follows. We will make the point clearer.
> > > >Another approach, such as ARDiT or Transfusion, repurposes the language model’s parameters for diffusion, leading to substantial computational demands.
> > > - The biggest advantage is that when combined with LLM or scaled up, DiTAR's three-stage approach (encode->LM->diffusion) can save a significant amount of computational load compared to ARDiT's single-stage approach(LM=diffusion).
> > > - Diffusion requires multiple sampling steps during inference. In models like ARDiT, where the LM and diffusion share parameters, each token prediction requires multiple computations on the LLM's parameters. In contrast, DiTAR only needs to perform multiple sampling steps on the diffusion head.
> > >
> > > \<Table1 The TFLOPs of generating a 10-second audio with NFE=10 and CFG>
> > > |Method|Parameters|TFLOPs↓|
> > > |--|--|--|
> > > |ARDiT|600M|8.70|
> > > |ARDiT|7B|112.78|
> > > |DiTAR(P=4)(Ours)|600M|2.75|
> > > |DiTAR(P=4)(Ours)|7B|5.40|
> > >
> > > 4. We are pleased to have addressed your concerns. We will further revise the paper,  adding more inference metrics and discussing more about the advantages and disadvantages of each system.
> > >
> > > 5. We are pleased to have addressed your concerns. We will make the experimental setup clearer in the revised manuscript.
> > >
> > > 6.  Comparison with VALLE-2：
> > >
> > > Thank you for the reminder. We followed the data processing methods mentioned in the VALL-E 2 paper.  All samples were evaluated using the same tool for WER/SIM. We have compiled all the results below for comparison. DiTAR matches VALL-E 2 in WER and significantly outperforms it in SIM.
> > >
> > > \<Table 2 LibriSpeech test-clean>
> > > |Method|Patch size|WER↓| SIM↑|
> > > |--|--|--|--|
> > > |GT |-|1.6|0.779|
> > > |VALL-E 2|1|**1.5**|0.643|
> > > |VALL-E 2|2|**1.5**|0.635|
> > > |VALL-E 2|4|2.2|0.615|
> > > |DiTAR (Ours)|1|4.65|0.694|
> > > |DiTAR (Ours)|2|1.55|**0.705**|
> > > |DiTAR (Ours)|4|**1.53**|0.678|
> > >
> > > 7. Comparison with ARDIT:
> > >
> > > Thank you for the reminder. We retrained our model using LibriTTS for 100k steps. All samples were evaluated using the same tool for WER/SIM. The results are shown below:
> > >
> > > \<Table 3 LibriTTS test-clean>
> > > |Method|WER↓| SIM↑|
> > > |--|--|--|
> > > |ARDiT|4.036|0.613|
> > > |DiTAR(Ours)|**3.536**|**0.615**|
> > >
> > > *We sincerely hope our reply addresses your concerns and that you might consider raising the rating.*

---

### Official Review · Reviewer_5BNx · 2025-03-13

**Overall Recommendation:** 3

**Summary:**

This paper introduces Diffusion Transformer Autoregressive Modeling (DiTAR), a patch-based framework that combines a language model with a diffusion process to generate continuous speech tokens. By employing a “divide-and-conquer” strategy, the language model processes aggregated patch embeddings, and the diffusion transformer then generates the next patch, reducing computational overhead and error compounding. Various experiments highlight the importance of each component and ultimately demonstrate that the methodology is both high-quality and efficient.

## update after rebuttal

The authors' rebuttal satisfactorily addressed most of my concerns through additional experiments and clarifications. The provided demos also demonstrated high quality. However, some ambiguity remains—specifically regarding how "low dim" is applied, as Figure 1 is still unclear on this point. I expect these issues to be clarified in the final manuscript. I am inclined to raise my score based on the improvements, but if the final version fails to resolve these outstanding issues, my score should remain unchanged.

**Claims And Evidence:**

Each claim is supported by the necessary results, and the experiments were conducted comprehensively and clearly.

**Essential References Not Discussed:**

The paper that originally introduced the definition of v-prediction should be mentioned [1].

[1] Salimans, T., & Ho, J. (2022). Progressive distillation for fast sampling of diffusion models. *arXiv preprint arXiv:2202.00512*.

**Experimental Designs Or Analyses:**

The necessary experiments are well-designed and the result analysis is convincing. The details of the aspect I wish to clarify through further queries are described in more detail below.

**Methods And Evaluation Criteria:**

Multiple models were evaluated using a unified benchmark, metric, and evaluation model, and the effort to maximize fairness was remarkable.

**Other Comments Or Suggestions:**

1. In Section 3.4, line 191, please confirm if “Eqn” should be replaced with “Eq”.
2. In Table 6, the System column shows “Seed-EN” twice; should the second occurrence be “Seed-ZH” instead?
3. For zero-shot TTS evaluation, it would be beneficial to add comparisons with the following recently proposed models:
    1. Multi-stage: E1 TTS [3]
    2. Single-stage: DiTTo-TTS [4]
    3. Open-sourced (production) models from TTS-Arena [5]—if possible, include comparisons with Kokoro, Fish Speech, XTTSv2, and StyleTTS2.
4. In Section 3.1.1, line 139, the statement “Noting the high similarity among adjacent continuous tokens, it is evident that a bidirectional dependency exists within local regions” would be strengthened by citing related research that supports this observation.
5. In Section 4.1.1, line 299, please consider citing the related works mentioned in “Many studies” to provide a more robust context for the discussion.

[3] Liu, Z., Wang, S., Zhu, P., Bi, M., & Li, H. (2025, April). E1 tts: Simple and fast non-autoregressive tts. In *ICASSP 2025-2025 IEEE International Conference on Acoustics, Speech and Signal Processing (ICASSP)* (pp. 1-5). IEEE.

[4] Lee, K., Kim, D. W., Kim, J., Chung, S., & Cho, J. DiTTo-TTS: Diffusion Transformers for Scalable Text-to-Speech without Domain-Specific Factors. In *The Thirteenth International Conference on Learning Representations*.

[5] https://huggingface.co/spaces/TTS-AGI/TTS-Arena

**Other Strengths And Weaknesses:**

Although this paper is not the first attempt to harness the strengths of LMs and diffusion models for creating efficient and high-performing generative models, the proposed approach is both valid and convincing, making its effective resolution of this challenge significant. Each claim is well-supported by appropriate experiments. I believe that this approach could serve as a general framework for generative modeling not only in speech synthesis but also in the image domain.

However, the following aspects require precise clarification.

1. The demo page and audio samples were not found; including them would further substantiate the evaluation results presented in the paper.
2. In the right-hand LocDiT diagram of Figure 1, is h_i included solely because of CFG? Must it be placed exactly there, and does the presence of h_i improve performance even without CFG—essentially, what is the quantitative contribution of h_i?
3. In Section 3.2 (line 171), does the term “lower-dimensional feature space” refer to h_i in Figure 1? Given that h_i is fed into LocDiT and should share the same hidden dimensions, its meaning in this context remains unclear.
4. In Section 3.3 (line 191), the notion of reduced “generality” is ambiguous—does this imply that the model can condition on its own self-supervised features rather than on externally labeled classes?
5. In Section 3.4 (line 183), v_θ is used without a formal definition; if it represents the v-prediction, please include a citation to its original introduction.
6. In Algorithm 1:
    1. On line 674, should “argmax” be replaced with “argmin”?
    2. On line 687, is “v” intended to be “v-hat”?
    3. The notation for tₙ within the for-loop requires clarification—for instance, on lines 685 and 687, does the resulting x correspond to x_t(n–1)?
7. In Section 3.5.1:
    1. The statement “the 24000Hz waveform is compressed into 40Hz latent with a dimension of 64” raises the question of whether training a simple VAE (as opposed to approaches like FSQ [2]) to compress raw audio to a 40Hz rate and 64-dimensional latent was challenging, and if any issues were encountered.
    2. Furthermore, since the codec’s reconstruction quality and code rate are critical hyperparameters for overall performance, it is important to know if ablation studies were conducted on these factors or if a performance comparison with other codecs was performed to clearly distinguish the contributions of the codec versus the language model.
8. In Section 3.5.4 regarding the prefix input to DiTAR’s LM:
    1. Is the ordering of inputs (prompting audio, text, target text) correct?
    2. If so, should prompting audio also be explicitly considered during training, as Figure 1 appears to suggest that inputs are fed in the order of (prompt) text, target text, and then prompting audio without separate training for prompting audio—this point requires clarification.
9. In Section 4.1.1 on Evaluation Metrics:
    1. For the Librispeech test-clean subset B, Faster-whisper-large-v3 was used; does this model guarantee performance comparable to OpenAI’s whisper-large-v3? If not, using a different evaluation measure for the F5 TTS subset B could compromise the fairness of the comparison.
    2. When measuring speaker similarity, was the ground truth raw waveform used instead of the codec-reconstructed audio?
10. In Section 4.2:
    1. Is the 20k to 280k hours of data used for scaling sourced from a different internal dataset than the training dataset mentioned in Section 4.1.1?
    2. Does the “Encoder” in Table 3 refer specifically to the aggregation encoder illustrated in Figure 1?
11. How long does it take for the evaluation model to converge during training?
12. In Section 4.3:
    1. Why is a patch size of 4 used instead of 2, given that Figure 3 indicates that a patch size of 2 achieves lower WER and higher similarity—could this decision be related to latency or throughput concerns?
    2. Does “historical context” refer to h_i or to historical patches, and in Table 4, does the value for Historical contexts represent the number of h_i’s or the number of patches?
    3. Compared to previous literature, DiTAR appears to perform well even with a extreamly low NFE (e.g., NFE = 2). Could the authors provide insights into the fundamental reason behind this phenomenon?
13. In Section 4.4.2, additional explanation regarding the NAR model would be beneficial—could the authors provide further details about its design and performance characteristics?

[2] Mentzer, F., Minnen, D., Agustsson, E., & Tschannen, M. (2023). Finite scalar quantization: Vq-vae made simple. *arXiv preprint arXiv:2309.15505*.

**Questions For Authors:**

1. Have you conducted any experiments in the image domain? Since the proposed method appears capable of addressing MAR’s challenges, it would be interesting to know how quality and latency improvements translate to image generation.
2. How does the proposed patchification differ from the approach in the MAR paper that generates multiple tokens simultaneously? Although including historical context sets them apart, it seems that fundamentally similar modeling is being employed.
3. Final feature matching is notoriously challenging, which is why many previous approaches adopt a coarse-to-fine strategy. However, DiTAR appears to achieve strong performance by aligning the final feature in one pass. Could the authors clarify whether this success is primarily attributable to the patchification strategy, or if there are other insights that explain this phenomenon?
4. In Section 3.5.4, the model is conditioned on phonemes; have you evaluated a character-based approach as well, and is there clear evidence that the phoneme-based conditioning offers distinct advantages?

**Relation To Broader Scientific Literature:**

As mentioned in the paper, the idea of combining the strengths of language models and diffusion for high-quality, efficient modeling in speech synthesis has been proposed recently. The significance here lies in demonstrating that this hybrid approach can outperform either modeling technique on its own. This finding is both meaningful and promising for future developments in the field.

**Theoretical Claims:**

The formulation was simple and convincing. However, one drawback is that the relationship between the learned v in Eq. 7’s flow-matching loss and the v-prediction in DDPM was not clearly explained. It would be beneficial to clarify this with a relevant citation.

---

> ### Author Rebuttal · Authors · 2025-03-31
>
> We appreciate your careful reading of our paper and your insightful comments. Due to the word limit, we reply point by point in a concise manner:
>
> **Questions in "Other Strengths And Weaknesses":**
>
> 1. Audio samples are presented here: https://spicyresearch.github.io/ditar/#hard-cases
> 2. No, $h_i$ it is not solely for CFG. $h_i$ serves as the output of the LM and the conditioning for LocDiT, connecting the two.
> 3. Yes. Compared with the total dim of a patch of tokens, the dim of $h$ is low.
> 4. Yes. We will replace the term "generality" with "generalization" to make the meaning clearer.
> 5. $v_t$ here represents vector field or velocity. We will add the description to reduce ambiguity.
> 6.  Thank you. We will correct the typos:
> 	1.  Line 674: argmax -> argmin
> 	2.  Line 687: $v$ -> $\hat{v}$
> 	3.  Line 685: $t_n$ -> $t_{n+1}$
> 7.  The implementation of VAE:
> 	1. Due to the lack of open-source speech VAE suitable for diffusion use, we trained the VAE following the approach used in LDM[1]. In the process, we aimed for adequate reconstruction quality without pursuing excessive compression, so the overall task was not challenging.
> 	2. Thank you for the suggestion. Extensive research on VAE is part of our future work.
> 8.  Prefix input to LM:
> 	1.  Thanks for pointing out the typo. We will fix it to (text, target text, prompting audio)
> 	2.  No. Following prior LM-based work [2], the loss is calculated over the entire audio.
> 9.  Evaluation Metrics:
> 	1.  To clarify, we've communicated with the authors of F5TTS, and they actually used 'faster-whisper-large-v3' instead of 'whisper-large-v3'. They admitted it was a typo in their paper. To maintain consistency, we continue to use "faster" in our paper.
> 	2.  Yes, follow prior works[2][3].
> 10. Section 4.2:
> 	1. 20k: part of Librilight, 60k: Librilight, 100k: Emilia, 280k: Librilight+Emilia+inhouse
> 	2. Yes.
> 11. Convergence time: 500k training steps (Appendix A.1).
> 12. Section 4.3:
>     1.  Yes. A patch size of 2 is slightly better than 4 in performance, but 4 offers better throughput. This section focuses on the ablation of other components, so any reasonable patch size, either 2 or 4, is OK.
>     2.  “Historical contexts” means the number of historical patches. We will make the expression clearer later on.
>     3.  Thank you for noticing this detail. We think the proposed historical patches and LM guidance enhance the accuracy of LocDiT's predictions.
> 13.  NAR in efficiency evaluation: It is a transformer identical to E2TTS[3]. It consists of 36 layers, each with a hidden dimension of 1024 and 16 heads. The performance is similar to E2TTS.
>
>
> **Questions in "Other Comments Or Suggestions":**
>
> 1&2. Thanks for pointing out the typo.
>
> 3. Other comparisons:
>
> 	- Kororo: does not support zero-shot mode.
>
> 	- StyleTTS 2 & E1TTS: tested on released subset of LibriTTS.
>
> 	- DiTToTTS: tested on Librispeech, but the subset used is not released.
>
> 	- XTTS v2 & Fish Speech: we evaluated the released checkpoints on Seed-EN.
> Test set| Method|WER↓| SIM↑|
> --|--|--|--|
> LibriTTS|StyleTTS 2|4.065|0.409|
> -|E1TTS|**3.246**|0.616|
> -|DiTAR(Ours)|3.401|**0.717**|
> LibriSpeech|DiTToTTS|2.56|0.627|
> -|DiTAR(Ours)|**1.78**|**0.64**|
> Seed-EN|XTTS v2|3.248|0.463|
> -|FishSpeech|2.372|0.55|
> -|DiTAR(Ours)|**1.685**|**0.735**|
>
> 4. Thank you. We will cite the corresponding reference[4].
> 5. Thank you for the suggestion, we will.
>
> **Questions in "Questions For Authors"**
>
> 1. Application on image generation is part of our future work.
> 2. The connection with MAR in patchification:
> 	- The purposes are different. MAR employs fully bidirectional attention and uses patchification to reduce computational demand. DiTAR is essentially a causal LM and uses patchification to perform bidirectional modeling locally within patches, which enhances performance.
> 	- The figure in the link better illustrates the differences: https://spicyresearch.github.io/ditar/#comparison
> 3. DiTAR's strong performance in one pass:  Patchification enables bidirectional modeling within patches, along with LM guidance, making predicting fine features more accurate. The LM->$h$->DiT->$x$ can be considered as an implicit coarse-to-fine process.
> 4. Phoneme vs. text: The use of phonemes is to align with other TTS systems for a fair comparison, not because phonemes are superior to text.
>
> **References**
>
> [1] Robin Rom., et al. High-Resolution Image Synthesis with Latent Diffusion Models, CVPR 2022
>
> [2] Wang, Chengyi, et al. "Neural codec language models are zero-shot text to speech synthesizers." _arXiv preprint_ (2023).
>
> [3] Eskimez, Sefik Emre, et al. "E2 tts: Embarrassingly easy fully non-autoregressive zero-shot tts." 2024 IEEE SLT. IEEE, 2024.
>
> [4] Ke. Tian, et al. Visual Autoregressive Modeling: Scalable Image Generation via Next-Scale Prediction, NeurIPS 2024
>
> *We sincerely hope that our reply could address your concerns and that you might consider raising the rating. Please let us know if you have any further questions or require additional results.*

---

> > ### Comment · Reviewer_5BNx · 2025-04-03
> >
> > I appreciate the authors’ rebuttal; their experiments and explanations have addressed most of my concerns. I also randomly listened to some of the demos provided and found the quality to be high. However, some ambiguity remains. For instance, it is still unclear whether "low dim" is concatenated with the patches or applied as a separate condition to the input, as its placement in Figure 1 suggests. I expect these issues, along with the promised revisions, to be clarified in the manuscript. Based on the improvements, I am willing to raise my score; however, if the final version does not resolve these issues satisfactorily, my score should remain at 2.

---

> > > ### Author Response · Authors · 2025-04-07
> > >
> > > We are pleased to have addressed most of your concerns and sincerely thank you for your constructive feedback. We promise to incorporate our discussions in the revised manuscript to reduce ambiguity.

---

### Official Review · Reviewer_oFFs · 2025-03-14

**Overall Recommendation:** 3

**Summary:**

This paper introduces DiTAR, a speech generation method that integrates a causal language model (LM) with a shallow diffusion module using a bidirectional diffusion transformer (LocDiT). The approach incorporates several key techniques, including patchifying continuous audio tokens, directly feeding historical patches into the diffusion module, classifier-free guidance for the diffusion model, and temperature-controlled ODE sampling to balance diversity and stability. Experimental results demonstrate state-of-the-art performance in zero-shot TTS, achieving strong robustness, speaker similarity, and high naturalness, while maintaining lower computational overhead compared to baseline models.


## update after rebuttal
I appreciate the authors' efforts in responding to my previous review, and I find that most of my concerns have now been effectively addressed. The work positions itself between fully autoregressive (AR) and non-autoregressive (NAR) methods. Its primary contribution appears to lie not in architectural novelty, but rather in its practical benefits and its good performance. I encourage the authors to incorporate the points discussed in the rebuttal, including relevant references, into the revised manuscript. Reflecting these improvements, I have increased my score from 2 to 3 and am now leaning towards acceptance.

**Claims And Evidence:**

The proposed methods and evaluation criteria are relevant.

**Essential References Not Discussed:**

Although this work should be considered a follow-up work to the prior work [1], the authors make only minimal references to it. The prior work explores the combination of a transformer and a shallow MLP decoder in both autoregressive and non-autoregressive settings, as well as the patchification of four tokens in continuous token experiments. Consequently, the contribution of this work is diminished unless the authors can demonstrate that the choice of autoregressive modeling is crucial rather than an arbitrary ordering.

[1] Fan, L., Li, T., Qin, S., Li, Y., Sun, C., Rubinstein, M., Sun, D., He, K., and Tian, Y. Fluid: Scaling autoregressive text-to-image generative models with continuous tokens. arXiv preprint arXiv:2410.13863, 2024.

**Experimental Designs Or Analyses:**

I reviewed the experimental design and analyses and found them sound.

**Methods And Evaluation Criteria:**

Proposed methods and/or evaluation criteria are reasonable and aligned with the problem/application.

**Other Comments Or Suggestions:**

I don't have any other comments or suggestions.

**Other Strengths And Weaknesses:**

Strengths:
* The strong performance in the main results demonstrates the effectiveness of the proposed modeling approach.
* The method is straightforward, and each component is systematically evaluated through ablation studies.

Weaknesses:
* The proposed method is closely related to Fluid [1], as both employ a language model backbone and a shallow diffusion module. The prior work also has shown autoregressive/random ordering in generation, patchfication of tokens. However, I do not see sufficient novelty, despite differences such as the use of a transformer in the shallow diffusion module, historical patching, diffusion module-only guidance, and temperature-based sampling for improving inference performance. This work should provide a comprehensive comparison with Fluid and clarify how it critically improves upon it.
* In the following sense, although the choice of autoregressive ordering is natural in TTS than non-autoregressive one with additional duration modeling, the authors should also justify the choice, as prior work has shown that random ordering can outperform autoregressive ordering. Additionally, since the model adopts autoregressive modeling, the paper should report inference speed metrics, including latency (time-to-first frame), full inference time, and the feasibility of real-time streaming.
* A more extensive ablation study would better highlight the importance of each module, including:
    * Comparing LocDiT to prior methods, such as the MLP-based diffusion module, and discussing whether its significant performance degradation in the absence of historical context stems from its transformer-based architecture rather than the MLP-based approach, given that prior works have demonstrated strong performance with MLP-based models.
    * Evaluating LM guidance, or diffusion module-only guidance, against prior methods, particularly those that apply guidance to the entire model using blank conditioning, while considering both performance and efficiency.
* The temperature-based sampling appears to have a marginal impact on performance.

[1] Fan, L., Li, T., Qin, S., Li, Y., Sun, C., Rubinstein, M., Sun, D., He, K., and Tian, Y. Fluid: Scaling autoregressive text-to-image generative models with continuous tokens. arXiv preprint arXiv:2410.13863, 2024.

**Questions For Authors:**

I don't have any other questions.

**Relation To Broader Scientific Literature:**

The key contributions to improve generation performance for TTS are applicable to broader audio generative modeling, including applications such as speech-language modeling and audio/music generation.

**Theoretical Claims:**

I reviewed proposed methods and found them sound.

---

> ### Author Rebuttal · Authors · 2025-03-31
>
> Thank you for your insightful comments. We provide detailed responses to your concerns as summarized below:
>
> Demo page: https://spicyresearch.github.io/ditar/#hard-cases
>
> **Q1. Fluid[1]  as an essential reference not discussed**
>
> To clarify, we have indeed been discussing MAR[2], the forerunner of AR with diffusion loss. Fluid and MAR are identical in methodology, with Fluid simply being a scaled version of MAR. Our discussion of MAR[2] essentially amounts to discussing Fluid. We will subsequently add references to Fluid for completeness.
>
> **Q2. Our contributions and connection with MAR/Fluid**
>
> Causal-attention AR (GPT-style) with a diffusion head performed poorly in predicting continuous tokens[1][2]:
> * MAR/Fluid abandoned the GPT style and proposed a bidirectional-attention method using random order.
> * Differently, we continue to delve deeper into GPT-style AR for continuous tokens, analyze why it does not perform well, and propose DiTAR as a solution.
>
> The figure in the link better illustrates the differences:： https://spicyresearch.github.io/ditar/#comparison
>
> **Q3. Different purposes of patchification**
>
> - The patchification technique is widely applied in various fields, but the main purpose is to reduce computational load and the best result is achieved when the patch size is set to 1[1][2][3][4].
> - Differently, in our work, the main purpose of patchification is to enable bidirectional modeling for next-patch prediction and overcome the limitations of causal AR. We achieve the best results when the patch size is greater than 1.
>
> **Q4.More inference metrics**
>
> Thank you for your suggestion. We have added more inference metrics. All metrics are obtained on an A100(80GB) GPU by generating 10-second audio.
>
> Batch size:500/1
> | Systems| Latency(s)↓| RTF↓| Throughputs(s)↑ |
> |---|---|--|--|
> | NAR|50.03/0.37|5.03/**0.037** |99.4/**27**|
> | DiTAR(P=4)|0.139/0.066|**1.39**/0.66|**360**/1.5|
> | DiTAR(P=2)|**0.1085**/**0.064** |2.17/1.28|230/0.78|
>
> As shown in the table, DiTAR's inference characteristics are similar to those of the causal language model. DiTAR has low latency and can use KV cache to save computation and increase concurrency, whereas NAR has high parallelism and can achieve fast speed with a small batch size.
>
> **Q5. The choice of Causal-AR  over NAR on speech generation**
>
> To clarify, the purpose of our paper is not to prove that AR models are more suitable for speech generation than NAR or other multi-stage methods, as each offers specific advantages and disadvantages depending on the scenario.
> Instead, our aim is to propose a *general GPT-style generative model based on continuous representations*, and to demonstrate its ability to achieve SOTA results when applied to speech generation. We kept the design minimalist and avoided domain-specific features like duration or prosody modeling, making it easier to scale and adapt to other generative fields, such as video and music.
>
> **Q6.LocDiT v.s. MLP, as the diffusion head**
>
> Actually, we have conducted the comparison in Table 4 of the paper. When the patch size is 1 and the number of historical patches is 0, LocDiT degrades to an MLP module. It is evident that LocDiT is significantly superior to the MLP. We will clarify this fact more clearly in the paper. [1][2] also demonstrated that causal AR with a MLP diffusion head performs poorly.
> | Method|WER↓| SIM↑|
> |----|-----|---|
> |MLP|53| 0.340|
> |LocDiT| **1.736** | **0.720** |
>
> **Q7.LM-guidance v.s. prior guidance methods for LM**
>
> Thank you for your suggestion. We further compare the proposed guidance method with the CFG method for language model[5].
> |Method|WER↓| SIM↑| Computational load↓|
> |----|:----|---|---|
> | Without any guidance| 2.858| 0.654| LM + diffusion |
> | CFG for language model [5]|2.323|0.680| LM x 2 + diffusion|
> | LM-guidance for LocDiT (ours)| **1.736**| **0.720** | LM + diffusion x 2|
>
> **Q8. Temperature for continuous-valued LM**
>
> The proposed temperature is aimed to balance diversity and certainty for continuous-valued LMs, not to improve WER/SIM. As demonstrated in Figure 6 of our paper, the higher the temperature, the greater the diversity in the generated results.
>
> **References:**
>
> [1] Lijie Li., et al. Fluid: Scaling autoregressive text-to-image generative models with continuous tokens. _arXiv preprint_ (2024).
>
> [2] Tianhong Li, et al. Autoregressive Image Generation without Vector Quantization. _arXiv preprint_ (2024).
>
> [3] Liu, Zhijun, et al. "Autoregressive diffusion transformer for text-to-speech synthesis." _arXiv preprint_ (2024).
>
> [4] Chen, Sanyuan, et al. "Vall-e 2: Neural codec language models are human parity zero-shot text to speech synthesizers." _arXiv preprint_ (2024).
>
> [5] Sanchez, Guillaume, et al. "Stay on topic with classifier-free guidance." _arXiv preprint_ (2024).
>
> *We sincerely hope our rebuttal addresses your concerns and that you might consider raising the rating. Please let us know if you have any further questions or require additional results.*

---

> > ### Comment · Reviewer_oFFs · 2025-04-07
> >
> > I appreciate the authors' efforts in responding to my previous review, and I find that most of my concerns have now been effectively addressed. The work positions itself between fully autoregressive (AR) and non-autoregressive (NAR) methods. Its primary contribution appears to lie not in architectural novelty, but rather in its practical benefits and its good performance.
> > I encourage the authors to incorporate the points discussed in the rebuttal, including relevant references, into the revised manuscript. Reflecting these improvements, I have increased my score from 2 to 3 and am now leaning towards acceptance.

---

> > > ### Author Response · Authors · 2025-04-07
> > >
> > > We are pleased to have addressed your concern. We sincerely appreciate your constructive feedback and raising the rating. We will incorporate your suggestions in the revised manuscript.

---

### Decision · Program_Chairs · 2025-05-01

**Decision:**

Accept (poster)

**Comment:**

Summary:
This paper introduces DiTAR, a novel architecture that combines the strengths of autoregressive language models (AR-LM) with diffusion transformers (DiT) for accurate and efficient zero-shot text-to-speech (TTS) generation using continuous tokens. The method breaks speech into patches, leverages a language model for long-term dependencies, and uses a Local Diffusion Transformer to generate local patches with bidirectional attention.

Key innovations:
- A patch-based divide-and-conquer architecture for managing continuous sequences.
- A novel temperature-controlled ODE-based sampling strategy.
- LM-guided classifier-free diffusion decoding.
- Competitive zero-shot TTS performance with significantly reduced computational overhead.

Strength:
- Novelty: Combines causal and bidirectional modeling in a novel way, tailored to continuous token modeling.
- Strong results: Outperforms strong baselines on multiple metrics with lower compute.
- Efficiency: Reduces function evaluations while maintaining quality.
- Scalability: Demonstrates clear scaling benefits across model and data sizes.
- Ablations: Systematic exploration of patch size, LM guidance, historical context, and temperature were done, validating design choices.

Weakness:
- Complexity: The multi-component design may be harder to reproduce or adapt by other technical teams.
- Multilingual support: While some multilingual results are reported, deeper analysis of multilingual issues is lacking.

Based on the reviewer feedback, the discussions, and AC assessment, this paper is recommended for Accept.